# SUMO is a pervasive regulator of meiosis

**Nikhil R Bhagwat[1,2], Shannon N Owens[2], Masaru Ito[1,2], Jay V Boinapalli[2], Philip Poa[2], Alexander Ditzel[2], Srujan Kopparapu[2], Meghan Mahalawat[2], Owen Richard Davies[3], Sean R Collins[2], Jeffrey R Johnson[4], Nevan J Krogan[4], Neil Hunter[1,2,5]\***

[1]Howard Hughes Medical Institute, University of California Davis, Davis, United States; [2]Department of Microbiology & Molecular Genetics, University of California Davis, Davis, United States; [3]Institute for Cell and Molecular Biosciences, University of Newcastle, Newcastle upon Tyne, United Kingdom; [4]Department of Cellular & Molecular Pharmacology, University of California San Francisco, San Francisco, United States; [5]Department of Molecular & Cellular Biology, University of California Davis, Davis, United States

**Abstract** Protein modification by SUMO helps orchestrate the elaborate events of meiosis to faithfully produce haploid gametes. To date, only a handful of meiotic SUMO targets have been identified. Here, we delineate a multidimensional SUMO-modified meiotic proteome in budding yeast, identifying 2747 conjugation sites in 775 targets, and defining their relative levels and dynamics. Modified sites cluster in disordered regions and only a minority match consensus motifs. Target identities and modification dynamics imply that SUMOylation regulates all levels of chromosome organization and each step of meiotic prophase I. Execution-point analysis confirms these inferences, revealing functions for SUMO in S-phase, the initiation of recombination, chromosome synapsis and crossing over. K15-linked SUMO chains become prominent as chromosomes synapse and recombine, consistent with roles in these processes. SUMO also modifies ubiquitin, forming hybrid oligomers with potential to modulate ubiquitin signaling. We conclude that SUMO plays diverse and unanticipated roles in regulating meiotic chromosome metabolism.

**\*For correspondence:**
nhunter@ucdavis.edu

**Competing interests:** The authors declare that no competing interests exist.

## Introduction

Meiosis precisely halves the chromosome complement enabling parents to contribute equally to their progeny while maintaining a stable ploidy through successive generations (*Hunter, 2015*). Ploidy reduction occurs by appending two rounds of chromosome segregation to a single round of replication to produce haploid gametes from diploid germline cells (*Figure 1A*). The key events of meiosis that ensure accurate segregation include the connection of homologous chromosomes by crossovers (*Hunter, 2006*), monopolar orientation of sister-kinetochores during meiosis I (*Watanabe, 2012*), and the stepwise loss of sister-chromatid cohesion, first from chromosome arms at anaphase I and then from sister centromeres at anaphase II (*McNicoll et al., 2013*). Crossover formation is the culmination of a complex series of interdependent chromosomal events during meiotic prophase I that include programmed homologous recombination, and the intimate pairing and synapsis of homologs (*Zickler and Kleckner, 2015*).

Meiotic recombination is initiated by programmed DNA double-strand breaks (DSBs), some 200–300 DSBs per nucleus in budding yeast, mouse, and human (*Lam and Keeney, 2015*). Ensuing recombinational interactions promote chromosome pairing and the assembly of synaptonemal complexes (SCs), densely packed transverse filaments with a zipper-like morphology that connect homologs during the pachytene stage (*Fraune et al., 2012*; *von Wettstein et al., 1984*; *Zickler and Kleckner, 1999*). Within the context of the SCs, selected recombinational interactions mature into

 

**eLife digest** Most mammalian, yeast and other eukaryote cells have two sets of chromosomes, one from each parent, which contain all the cell's DNA. Sex cells – like the sperm and egg – however, have half the number of chromosomes and are formed by a specialized type of cell division known as meiosis. At the start of meiosis, each cell replicates its chromosomes so that it has twice the amount of DNA. The cell then undergoes two rounds of division to form sex cells which each contain only one set of chromosomes. Before the cell divides, the two duplicated sets of chromosomes pair up and swap sections of their DNA. This exchange allows each new sex cell to have a unique combination of DNA, resulting in offspring that are genetically distinct from their parents.

This complex series of events is tightly regulated, in part, by a protein called the 'small ubiquitin-like modifier' (or SUMO for short), which attaches itself to other proteins and modifies their behavior. This process, known as SUMOylation, can affect a protein's stability, where it is located in the cell and how it interacts with other proteins. However, despite SUMO being known as a key regulator of meiosis, only a handful of its protein targets have been identified.

To gain a better understanding of what SUMO does during meiosis, Bhagwat et al. set out to find which proteins are targeted by SUMO in budding yeast and to map the specific sites of modification. The experiments identified 2,747 different sites on 775 different proteins, suggesting that SUMO regulates all aspects of meiosis. Consistently, inactivating SUMOylation at different times revealed SUMO plays a role at every stage of meiosis, including the replication of DNA and the exchanges between chromosomes. In depth analysis of the targeted proteins also revealed that SUMOylation targets different groups of proteins at different stages of meiosis and interacts with other protein modifications, including the ubiquitin system which tags proteins for destruction.

The data gathered by Bhagwat et al. provide a starting point for future research into precisely how SUMO proteins control meiosis in yeast and other organisms. In humans, errors in meiosis are the leading cause of pregnancy loss and congenital diseases. Most of the proteins identified as SUMO targets in budding yeast are also present in humans. So, this research could provide a platform for medical advances in the future. The next step is to study mammalian models, such as mice, to confirm that the regulation of meiosis by SUMO is the same in mammals as in yeast.

crossovers such that each pair of chromosomes attains at least one crossover despite a low number of events per nucleus. Homologs then desynapse and prepare for the meiosis-I division. The connections created by crossovers enable the stable bipolar orientation of homologs on the meiosis-I spindle, and thus accurate segregation during meiosis I. By creating new combinations of gene alleles, crossing over and independent chromosome segregation during meiosis fuels natural selection.

Orchestrating the elaborate events of meiosis are regulatory networks that function at the transcriptional, post-transcriptional, translational, and post-translational levels (*Bose et al., 2014*; *Brar et al., 2012*; *Cahoon and Hawley, 2016*; *Cheng et al., 2018*; *Crichton et al., 2014*; *Gao and Colaiácovo, 2018*; *Govin and Berger, 2009*; *Gray and Cohen, 2016*; *Jin and Neiman, 2016*; *Nottke et al., 2017*; *Otto and Brar, 2018*; *Tresenrider and Ünal, 2018*). The post-translational, SUMO (Small Ubiquitin-like MOdifier) protein-modification system (SMS) is now recognized as an essential regulator of meiotic prophase (*Cheng et al., 2007*; *de Carvalho and Colaiácovo, 2006*; *Lake and Hawley, 2013*; *Nottke et al., 2017*; *Rodriguez and Pangas, 2016*; *Sakaguchi et al., 2007*; *Vujin and Zetka, 2017*; *Watts and Hoffmann, 2011*). Like ubiquitin, SUMO is conjugated to lysine (K) side-chains on target proteins via a cascade of enzymes that activate (E1) and conjugate (E2) SUMO, and provide target specificity (E3 ligases) (*Jürgen Dohmen, 2004*; *Gareau and Lima, 2010*; *Jentsch and Psakhye, 2013*; *Johnson, 2004*; *Zhao, 2018*). SUMOylation can also be reversed by the action of dedicated proteases (*Kunz et al., 2018*). The consequences of SUMOylation are varied and target specific (*Zhao, 2018*), but include conformational changes, creating and masking binding interfaces to mediate protein interactions, and competing with other lysine modifications such as ubiquitylation and acetylation (*Almedawar et al., 2012*; *Flotho and Melchior, 2013*; *Liebelt and Vertegaal, 2016*; *Papouli et al., 2005*; *Steinacher and Schär, 2005*).

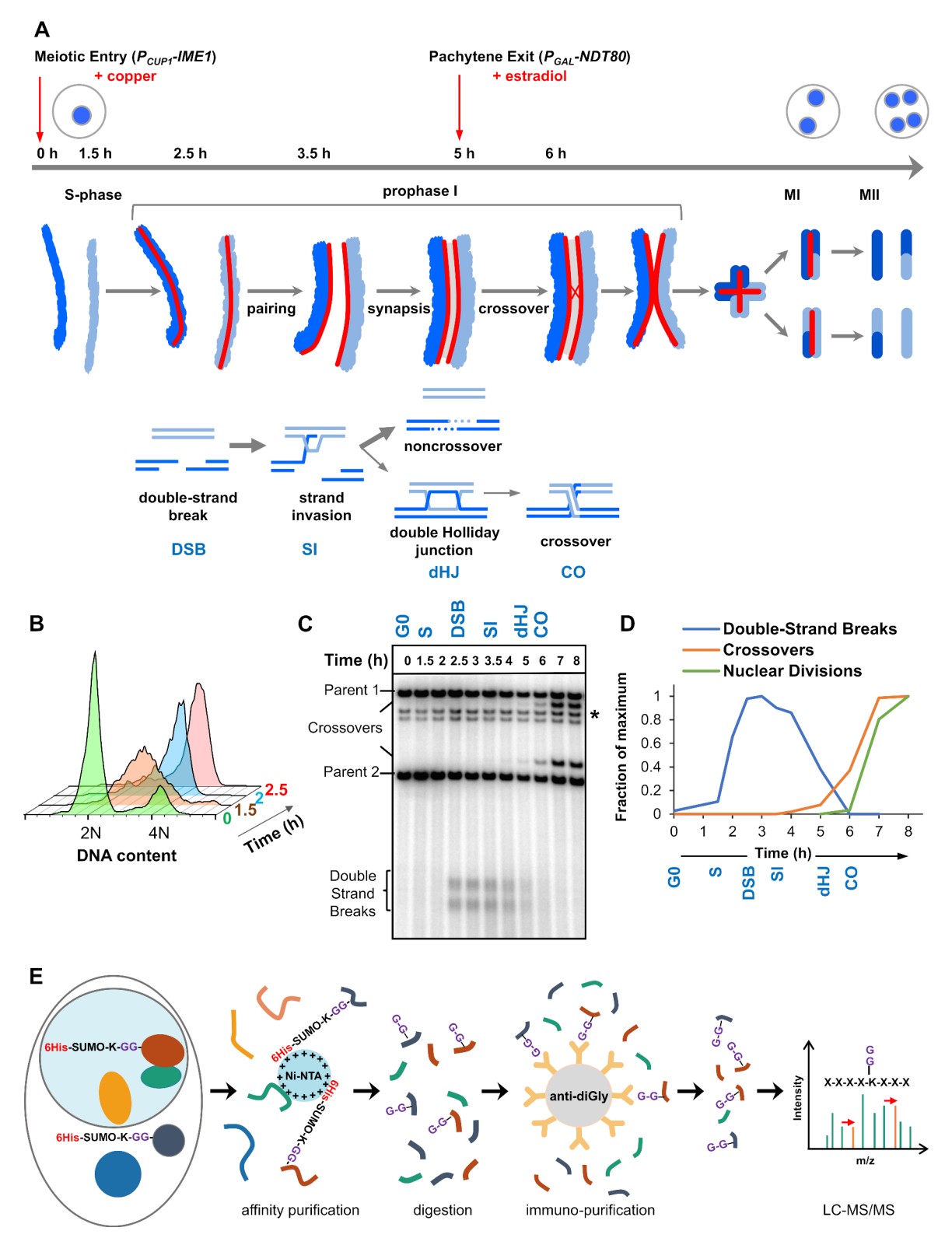

**Figure 1.** Experimental approach for profiling SUMOylation during meiosis. (**A**) Cell synchronization regimen showing the timing of nuclear (top), chromosomal (middle) and recombination (bottom) events. Samples were collected at the indicated timepoints (h, hours). Red arrows denote induction points of *IME1* and *NDT80* expression. (**B**) Flow cytometry data illustrating the progression of meiotic S-phase following $P_{CUP1}$-*IME1* induction. (**C**) Southern blot image showing the progression of recombination at the *HIS4::LEU2* hotspot. Asterisk indicates cross-hydridizing bands. (**D**) Timing and

*Figure 1 continued on next page*

*Figure 1 continued*

synchrony of meiotic cultures. Levels of DSBs and crossovers at *HIS4::LEU2*, and nuclear divisions (MI±MII) were normalized to 1. The timing of SEIs and dHJs was determined using 2D gels (*Figure 1—figure supplement 1*). (E) Regimen for purification of peptides harboring K-ε-GG SUMO remnants (G-G-). (Ni-NTA, nickel-nitrilotriacetic acid resin; anti-diGly, anti-K-ε-GG antibody beads).

The online version of this article includes the following figure supplement(s) for figure 1:

**Figure supplement 1.** Two-dimensional Southern blot analysis of joint-molecule recombination intermediates.

To date, only a handful of meiotic SUMO conjugates have been identified and studied in any detail. In budding yeast, these include the SC component Ecm11 (*Humphryes et al., 2013*; *Zavec et al., 2008*), SUMO E2 conjugase, Ubc9 (*Klug et al., 2013*), core recombination factor, Rad52 (*Sacher et al., 2006*), chromosome-axis protein Red1 (*Cheng et al., 2013*; *Eichinger and Jentsch, 2010*; *Lin et al., 2010*; *Zhang et al., 2014*), and type-II topoisomerase Top2 (*Zhang et al., 2014*). Also, in *Caenorhabditis elegans*, SUMO targets components of the chromosome congression/segregation ring complexes (*Davis-Roca et al., 2018*; *Pelisch et al., 2017*).

This paucity of examples underscores how understanding meiotic SUMOylation has been impeded by inefficient piecemeal approaches to identifying targets and mapping the sites of SUMO conjugation. To overcome this impediment, we developed an efficient proteomics regimen to map SUMO-conjugation sites proteome-wide during meiosis in budding yeast. In combination with label-free quantitation (LFQ) and highly synchronous meiotic time courses, this approach allows SUMOylation at protein and site levels to be monitored during the key transitions of homologous recombination, chromosome pairing and synapsis. The resulting mass spectrometry (MS) datasets provide a comprehensive and unprecedented view of the SUMO landscape, revealing dynamic waves of modification coincident with the major events of meiotic prophase I. Functional classes of SUMO targets imply roles in basic cellular functions including metabolism, chromatin organization, transcription, ribosome biogenesis, and translation. In addition, meiosis-specific aspects of chromosome metabolism are strongly represented pointing to roles for SUMO in regulating recombination, chromosome synapsis, and segregation. These inferences were explored by acutely inactivating de novo SUMOylation at different times during meiotic prophase. This analysis reveals distinct execution points for SUMO modification and identifies roles in the onset of S-phase, DSB formation, crossing over, and chromosome synapsis. Together, our analysis delineates a diverse and dynamic meiotic SUMO-modified proteome and provides a rich resource toward a mechanistic understanding of how SUMO regulates the complex events of meiosis.

## Results and discussion

### Cell synchronization and purification of SUMO-Modified peptides

Standard meiotic time-courses in budding yeast have relatively poor temporal resolution of the key events of meiotic prophase I. To sharpen culture synchrony, we employed the method of Berchowitz et al. in which cells synchronize in G0 before meiosis is triggered by inducing expression of the master regulator, *IME1*, which is under control of the copper-inducible *CUP1* promoter ($P_{CUP1}$-*IME1*; *Figure 1A*; *Berchowitz et al., 2013*). Five hr after induction of meiosis, cells were synchronized for a second time during pachytene, when homologs are fully synapsed. This was achieved by reversibly arresting cells using an estradiol inducible *NDT80* gene ($P_{GAL}$-*NDT80*) (*Benjamin et al., 2003*), which encodes a meiosis-specific transcription factor required for progression beyond pachytene. Upon $P_{GAL}$-*NDT80* expression, double-Holliday junction intermediates (dHJs) are rapidly resolved into crossovers, SCs disassemble and cells progress to MI (*Allers and Lichten, 2001*; *Chu and Herskowitz, 1998*; *Clyne et al., 2003*; *Sourirajan and Lichten, 2008*).

Cell samples from synchronized cultures were harvested and processed for SUMO proteomics at six different time-points that capture the key transitions of meiotic prophase I (*Figure 1A–D* and *Figure 1—figure supplement 1*). Cells prior to $P_{CUP1}$-*IME1* induction were sampled as a pre-meiotic control (G0). 1.5 hr after $P_{CUP1}$-*IME1* induction (**S**), cells were in meiotic S-phase, but DSB formation had not begun. By 2.5 hr (**DSB**), DNA replication was complete and DSB formation was ongoing. 3.5 hr (**SI**), captures the events of DNA strand-invasion and accompanying SC formation. By 5 hr (**dHJ**), cells were arrested in pachytene with fully synapsed chromosomes and unresolved dHJs. $P_{GAL}$-

*NDT80* expression was then induced and cells harvested 1 hr later (**CO**), as dHJs were being resolved into crossovers but meiotic divisions had not yet begun.

To obtain the highest quality peptide samples for SUMO proteomics, we addressed three major impediments: (i) proteases are hyper-activated in meiotic cells (*Klar and Halvorson, 1975*); (ii) the stoichiometry of SUMOylation is typically very low (the 'SUMO paradox') (*Hay, 2005*); (iii) the native branched SUMO remnant produced by trypsin digestion (K-ε-GGIQE) is not amenable to efficient MS-based identification (*Wohlschlegel et al., 2006*). Thus, strains were generated in which the native *SMT3* locus was engineered to express hexa-histidine tagged Smt3 with an I96K mutation (6His-Smt3-I96K; *Figure 1E*; functionality of this construct is reflected in the 96% spore viability of *6His-Smt3-I96K/6His-Smt3-I96K* homozygotes and the timing and efficiency of meiosis in these strains, *Figure 1B–D* and *Figure 1—figure supplement 1*; *Tammsalu et al., 2015*; *Wohlschlegel et al., 2006*; *Xu et al., 2010*). This construct enabled a two-step purification regimen to yield samples that were highly enriched for peptides with a K-ε-GG di-glycine SUMO-conjugation remnant, which is readily detected by tandem MS. SUMO-conjugated proteins were initially enriched using immobilized metal-affinity chromatography under denaturing conditions (6M guanidine), thereby limiting proteolysis. Eluted samples were then split and digested with LysC, or a combination of LysC plus GluC, to yield peptides with di-glycine branched SUMO-remnants that are amenable to further affinity purification using anti-di-glycyl-lysine antibodies (*Xu et al., 2010*). Eluted peptides were subjected to LC-MS/MS over a 90 min acetonitrile gradient on a Q-Exactive Orbitrap (Thermo Scientific) with data-dependent acquisition; and data were processed using MaxQuant and Perseus software (Max Planck Institute) (*Cox et al., 2014*; *Cox and Mann, 2008*; *Tyanova et al., 2016*). Critically, this method unambiguously distinguishes between conjugation sites for SUMO and ubiquitin because LysC (± GluC) digestion does not yield a di-glycine remnant for ubiquitin (for ubiquitin, K-ε-GG is a product only when trypsin is employed; *Tammsalu et al., 2015*; *Wohlschlegel et al., 2006*; *Xu et al., 2010*). To obtain biological replicates with high correlation scores ($r \geq 0.8$ Pearson correlations, *Figure 2—figure supplement 1A*) for quantitative analysis of SUMOylation dynamics, samples were collected from three independent time courses, processed in parallel, and then LC-MS/MS was performed in tandem with a randomized sample order and identical run conditions. In order to maximize the identification of target proteins and their conjugation sites, data from additional time courses were included in a separate analysis.

## Key features of the SUMO-modified proteome during meiotic prophase I

Parallel analysis of samples digested separately with LysC and LyC + GluC enhanced peptide coverage, thereby increasing the number of SUMOylated proteins identified by 11% and the number of conjugation sites mapped by 27%. Combined analysis of all samples identified 2747 SUMOylation sites in 775 proteins. By comparison, the most comprehensive analysis of SUMO targets in vegetative yeast to date identified 244 targets and 257 sites (*Esteras et al., 2017*). Of these targets, 166 were also identified in our analysis indicating that meiotic and mitotic SUMO targets show some overlap (*Supplementary file 1* summarizes previous yeast SUMO MS/MS studies in vegetative cells).

For individual time-point samples in the three parallel time courses, between 505 and 553 target proteins, and 1147 and 1362 sites were identified, with a majority of proteins being identified in multiple time points (*Figure 2A and B*). The relatively narrow numerical range of proteins identified across samples is one indication that sample preparation was consistent over the entire set. Also, of the 2414 conjugation sites identified in a single MaxQuant run, 1866 could be assigned with high confidence, with localization probabilities of $\geq 0.96$ (*Figure 2C*, *Supplementary file 2*). Moreover, a majority of lower confidence sites remain viable for functional analysis because decreased probabilities typically stemmed from ambiguity between two adjacent lysines in the same peptide.

Ubc9 binds directly to consensus SUMOylation peptide, ψ-K-x-E/D (ψ = large hydrophobic residue), to favor modification at these sites (*Bernier-Villamor et al., 2002*; *Rodriguez et al., 2001*; *Sampson et al., 2001*). However, only 14.26% of identified conjugation sites conformed to this consensus, with an additional 0.50% displaying the hydrophobic variant (ψψψ -K-x-E/D) and 3.23% having the reverse consensus sequence (E/D-x-K-ψ; *Figure 2D*). Around a fifth of sites comprised partial-consensus acidic (K-x-E/D, 11.57%) and reverse-acidic (E/D-x-K, 10.32%) motifs. Di-lysine was the only other recognizable motif (7.55%). Thus, a majority of modified sites (52.57%) did not have a recognizable motif. Identified sites were compared to those predicted by GPS-SUMO software

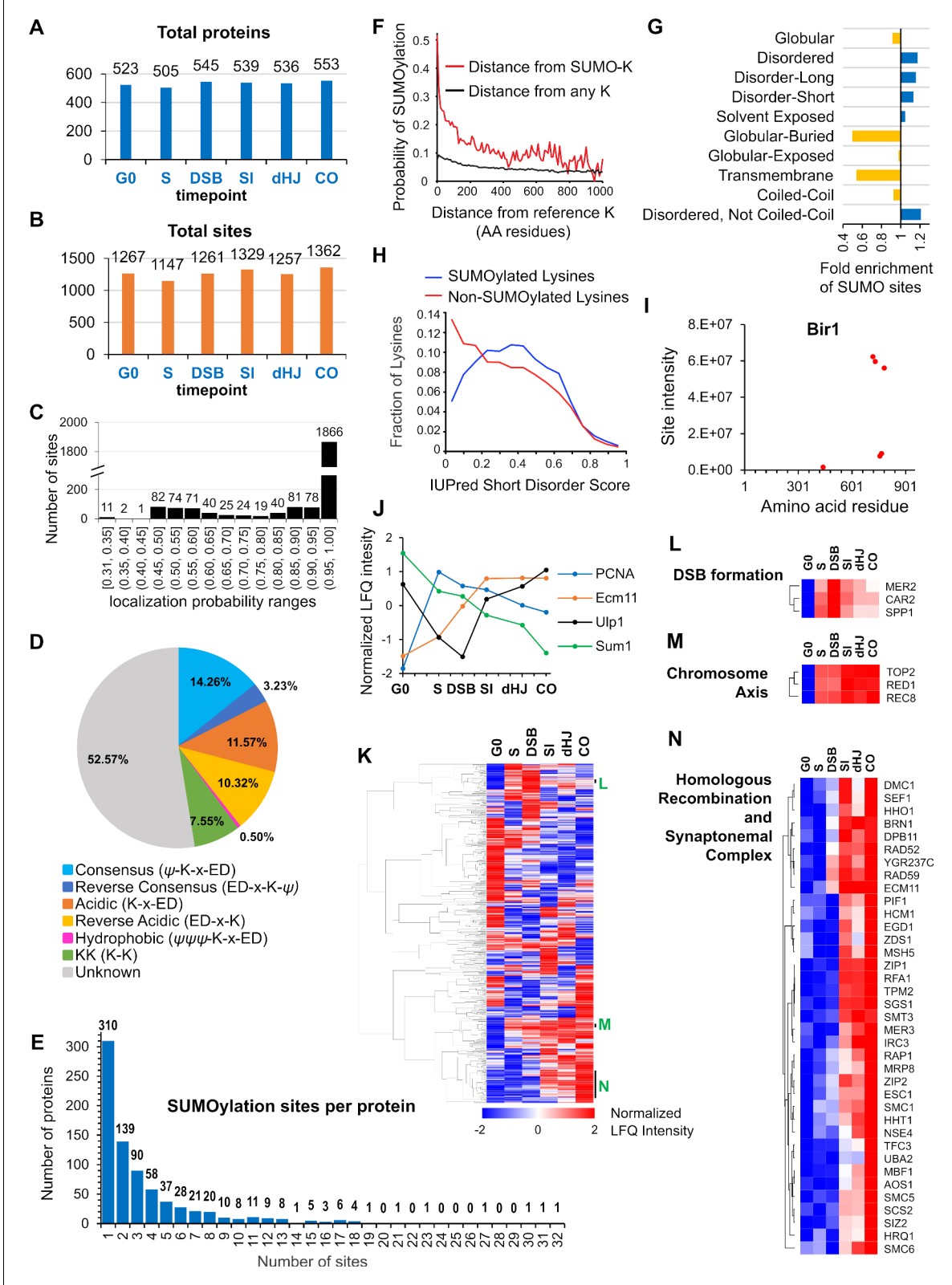

**Figure 2.** Characteristics of SUMOylation sites and temporal LFQ profiles of SUMOylated proteins. (**A-B**) Total numbers of proteins (**A**) and SUMO sites (**B**) identified from each time point in the triplicate experiments used for quantitative analysis. (**C**) Localization probabilities for identified SUMO sites as calculated by MaxQuant. (**D**) Proportions of identified SUMOylation sites that conform to indicated consensus sequences. Ψ, hydrophobic amino acid; x: any amino acid. (**E**) Distribution of SUMOylation sites per protein. (**F**) Probability of a lysine being SUMOylated as a function of its distance from

*Figure 2 continued on next page*

*Figure 2 continued*

either a SUMOylated lysine (red line) or any lysine (black line). (**G**) Enrichment or depletion of SUMOylation sites relative to protein secondary structure. (**H**) Distributions of all lysines (red line) and SUMOylated lysines (blue line) relative to IUPred short disorder score. (**I**) Site intensity profile of chromosomal passenger complex component Bir1 illustrating site clustering. (**J**) Normalized temporal LFQ profiles for PCNA, Ecm11, Ulp1 and Sum1 highlighting the diverse dynamics of SUMOylation during meiosis. (**K**) Hierarchical clustering of normalized LFQ profiles. (**L**) (**M**) (**N**) Clustering of SUMO targets involved in (**L**) DSB formation, (**M**) chromosome axes, and (**N**) recombination and synapsis.

The online version of this article includes the following figure supplement(s) for figure 2:

**Figure supplement 1.** Characteristics of the the meiotic SUMO-modified proteome.
**Figure supplement 2.** Western blot validation of SUMO targets.
**Figure supplement 3.** Ribosomal protein network.

---

(*Zhao et al., 2014*). Even when the search threshold setting was 'low', less than 19% (512 of 2745) of identified sites were predicted, emphasizing the limited utility of such software and the value of our proteomics dataset (*Supplementary file 3*).

Two or more conjugation sites were identified in 465 (60%) of the 775 identified SUMOylated proteins and the distribution of site numbers had a long tail, with 178 (23%) proteins containing five or more sites and three proteins with ≥30 sites (Ulp1, Red1 and Sir4; *Figure 2E*). When multiple conjugation sites were present in a single protein, they tended to cluster, with a 47% probability of adjacent SUMO sites being less than five residues apart (*Figure 2F*). This clustering effect implies that target features such as local secondary structure, solvent exposure and targeting by E3 ligases are more important determinants of SUMOylation site specificity than the presence a consensus conjugation motif, which is absent from a majority of sites. Consistently, SUMO sites were depleted from regions of predicted globular, buried or transmembrane structure, and were enriched in regions of moderate disorder (*Figure 2G and H*). For each target protein identified in our analysis, we generated a diagram detailing (i) the locations of SUMO sites relative to non-SUMOylated lysines, (ii) predicted SIMs, (iii) PFAM domains, and (iv) protein secondary structure (*Supplementary file 5*).

## SUMOylation dynamics and target identities

### SUMOylation site intensities and dynamics

Label-free quantification (LFQ) (*Cox et al., 2014*) further betrayed the immense complexity of meiotic SUMOylation (*Figure 2I–N*). First, for individual targets with multiple conjugation sites, cumulative site intensities (across all time points) provides a readout of relative site usage and site clustering (*Figure 2I*; also see examples below). This analysis also has predictive value for identifying functionally important sites, as exemplified by the much higher intensities of sites with known physiological roles in the SUMO E2 ligase, Ubc9 (K153, Figure 4C), and the SC component Ecm11 (K5, Figure 5M), and discussed further below; (*Klug et al., 2013*; *Leung et al., 2015*). Second, intensity profiles revealed that different sites within a single target can have distinct SUMOylation dynamics, for example, the profile of histone H4-K77 is distinct from those of the other SUMOylated H4 sites (Figure 7F). Third, normalized LFQ profiles showed that different targets can have radically different SUMOylation dynamics (*Figure 2J*). However, hierarchical analysis of LFQ profiles rendered clusters of functionally related proteins (e.g. DSB formation and HR), and physically interacting proteins (e.g. chromosome axis and SC) (*Figure 2K–N*). SUMOylation profiles are a compound readout of changes in SUMO modifications and protein levels. The SUMOylation profiles of selected conjugates were validated by western blot (*Figure 2—figure supplement 2*); in general, the results matched the proteomics profiles implying that the inferred dynamics tend to reflect changes in target SUMOylation and not simply protein levels (*Figure 2—figure supplement 2, A–C*). Red1 was a notable exception for which relative SUMOylation levels appeared more or less constant, with changes primarily reflecting the total protein level, which increased throughout prophase I (*Figure 2—figure supplement 2D*).

### Gene ontology and network analysis

Gene ontology (GO) analysis of targets showed strong enrichment for nuclear processes (*Figure 2—figure supplement 1B and C*), especially those associated with chromosome metabolism and ribosome biogenesis, as seen for mitotically cycling cells (*Albuquerque et al., 2013*;

*Albuquerque et al., 2015*; *Esteras et al., 2017*; *Hendriks et al., 2014*; *Srikumar et al., 2013a*). However, cytosolic processes were also enriched due the strong representation of glycolytic enzymes and other carbohydrate metabolic processes. Membrane-associated processes were notably underrepresented, although this could in part reflect their low expression, hydrophobicity and lower density of Lys-C cutting sites (*Vit and Petrak, 2017*). Network analysis revealed large clusters of physically interacting proteins consistent with the group SUMOylation paradigm advanced by Jentsch and colleagues (*Jentsch and Psakhye, 2013*; *Psakhye and Jentsch, 2012*). The most striking example is the ribosome with almost all of the 40S and 60S subunits being SUMOylated, for a total of 84 proteins and 527 conjugation sites (*Figure 2—figure supplement 3* and *Supplementary file 4*).

## SUMOylation is required throughout meiotic prophase I

The diversity of targets identified by our analysis points to roles for SUMOylation throughout meiotic prophase I and at each step of HR. To test this assertion and define execution points for SUMOylation, we employed an auxin-induced degron (AID) allele of the E1 subunit Aos1 to acutely block de novo SUMOylation at four key transition points (*Figure 3A,B* and *Figure 3—figure supplement 1*; very similar results were obtained using a Uba2-AID degron allele, *Figure 3—figure supplements 2* and *3*). In each case, meiotic cultures were split at the appropriate time point, and auxin was added to one sub-culture to induce degradation of Aos1-AID with the other sub-culture serving as a positive control. When Aos1-AID degradation was induced 30 min after entry into meiosis (experiment 1, *Figure 3B and C*), the onset of S-phase, as assessed by FACS analysis, was delayed by ≥30 min and meiotic divisions were completely blocked (*Figure 3D and E*). Identified SUMOylation targets involved in DNA replication, cell cycle, and metabolic regulation could be responsible for these phenotypes (*Supplementary file 4*).

To uncover post S-phase functions of SUMO that could account for the block to meiotic divisions, cells were synchronized using an analog-sensitive allele of the Cdc7 kinase (*cdc7-as*; condition 2, *Figure 3F,G*; *Wan et al., 2006*). Treatment of *cdc7-as* cells with the ATP analog PP1 causes meiotic cultures to reversibly arrest after S-phase, but prior to the initiation of recombination by DSB formation. The DNA events of HR were monitored using Southern blot assays at a well-characterized DSB hotspot (*Figure 3—figure supplement 1A–E*); and synapsis was analyzed by immunostaining chromosome spreads for the major SC component Zip1 (*Figure 3—figure supplement 1F*). Degradation of Aos1-AID immediately following release from *cdc7-as* arrest blocked DSB formation for ~2 hr indicating an unanticipated role for SUMO in the initiation of HR. Identified targets that could affect DSB formation include cohesin, Hop1, Red1, Spp1, Mer2, Rec114, and Dbf4 (discussed below; *Supplementary file 4*).

Condition 2 also resulted in a complete block to crossing over and meiotic divisions (*Figure 3G*), suggesting additional roles for SUMO in HR and/or the progression of meiosis. Therefore, we also determined the effects of degrading Aos1-AID later, just after DSBs were formed following release from *cdc7-as* arrest (condition 3, *Figure 3H,I*). In control cells (no auxin), initial DSB levels were high and continued to rise for 1 hr, before being repaired to yield high levels of crossovers and noncrossovers. When Aos1-AID was degraded, initial DSB levels were also high, but instead of continuing to rise, levels immediately decreased. These observations are consistent with the role for SUMO in DSB formation defined above, with later degradation of Aos1-AID preventing only late-forming DSBs. Joint molecule (JM) strand-exchange intermediates appeared to form efficiently following Aos1-AID degradation but reached peak levels with a ~ 1 hr delay relative to control cells suggesting slower formation. Subsequent resolution of JMs was severely delayed and cells again failed to divide. Consistent with a JM resolution defect, crossover and non-crossover products were reduced by 70% and 48%, respectively (*Figure 3I*).

For condition 3, we also analyzed chromosome synapsis by immunostaining for the major SC component Zip1 and quantifying four classes of nuclei: no staining, foci only, partial synapsis with both lines and foci of Zip1, and full synapsis with extensive linear staining (*Figure 3J* and *Figure 3—figure supplement 1F*). Both synapsis and, unexpectedly, de-synapsis were defective when Aos1-AID was degraded. Without auxin, synapsis levels peaked at 10 hr (1 hr after the culture was split), with partial synapsis in 53% of nuclei and full SCs in 19%. De-synapsis rapidly ensued and by 11 hr, 75% of cells had no Zip1 staining. By comparison, Aos1-AID degradation appeared to stall synapsis, with levels remaining largely unchanged between 9 and 10 hr (p=0.09, *G*-test; *Figure 3J*). However,

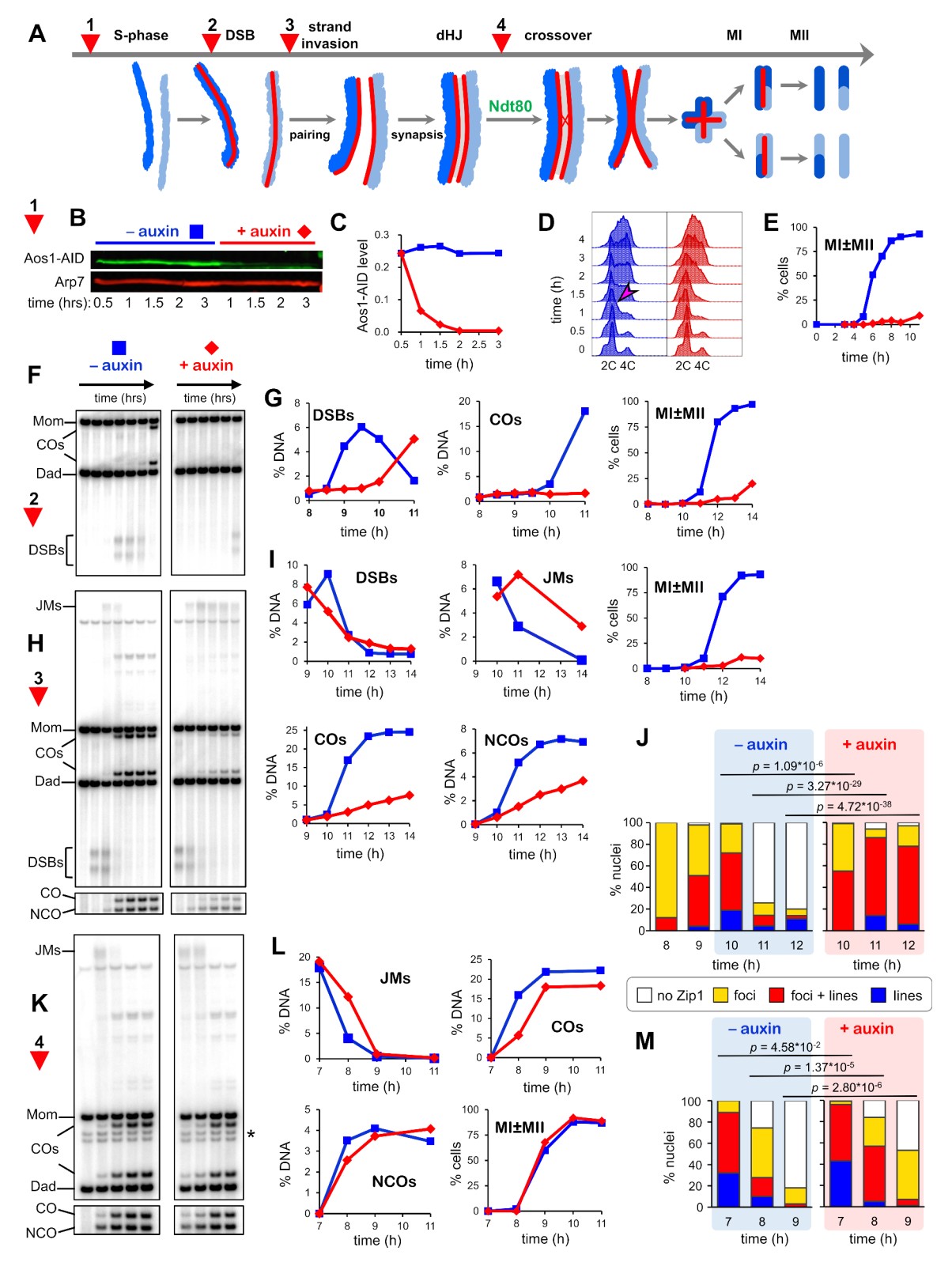

**Figure 3.** SUMOylation functions throughout meiosis. (**A**) Cartoon showing the four points (experiments 1–4) at which de novo SUMOylation was acutely inactivated. (**B**) Immunoblot of Aos1-AID with and without the addition of auxin at 30 mins (experiment 1). Arp7 was used as loading control. (**C**) Quantification of the blot shown in (**B**). (**D**) Flow cytometry analysis of S-phase progression for experiment 1. The magenta arrow indicates the onset of S-phase in control cells. (**E**) Meiotic nuclear divisions (MI ± MII) for experiment 1. (**F**) 1D gel Southern blot image for analysis of DSBs and crossovers in

*Figure 3 continued on next page*

*Figure 3 continued*

experiment 2. (G) Quantification of DSBs, crossovers and meiotic divisions for experiment 2. (H) 1D gel Southern blot images for experiment 3. The top gel was used to quantify DSBs and crossovers (COs). The bottom gel was used to quantify non-crossover products (NCOs). (I) Quantification of DSBs, joint molecules (JMs), COs, NCOs, and meiotic divisions for experiment 3. JMs were quantified from 2D gel Southern analysis (*Figure 3—figure supplement 4*). (J) Quantification of synapsis (Zip1-staining classes) for experiment 3 (*Figure 3—figure supplement 1F*). (K) 1D gel Southern blot images for experiment 4. The top gel was used to quantify DSBs and crossovers (COs). The bottom gel was used to quantify non-crossovers (NCOs). (L) Quantification of DSBs, joint molecules (JMs), COs, NCOs and meiotic divisions for experiment 4. JMs were quantified from 2D gel Southern analysis (*Figure 3—figure supplement 4*). (M) Quantification of synapsis (Zip1-staining classes) for experiment 4.

The online version of this article includes the following figure supplement(s) for figure 3:

**Figure supplement 1.** Analysis of meiotic recombination and synapsis.
**Figure supplement 2.** SUMO execution-point analysis using the Uba2-AID degron allele.
**Figure supplement 3.** Degron-mediated depletion of Aos1 and Uba2.
**Figure supplement 4.** 2D Southern blot analysis of joint molecule recombination intermediates.

by 11 hr, very high levels of synapsis were achieved; 72% of cells had partial synapsis and 14% had full SCs. At 12 hr, synapsis levels were almost unchanged suggesting defective de-synapsis. Thus, de novo SUMOylation has both a post-DSB function to promote the timely formation of JMs and SCs, and a post-synapsis function in JM resolution and de-synapsis. These roles of SUMO may be mediated by the numerous targets identified in processes such as HR, SC formation, the DNA damage response, and transcription (discussed below in Figures 5-7; *Supplementary file 4*).

Finally, late roles of de novo SUMOylation were determined by degrading Aos1-AID as cells were released form pachytene arrest (condition 4; *Figure 3A*) using the $P_{GAL}$-NDT80 allele (see above and *Figure 1A*; *Benjamin et al., 2003*). In contrast to the other conditions, meiotic divisions occurred efficiently when Aos1-AID was degraded at this late stage, suggesting that de novo SUMOylation is not essential for MI and MII (*Figure 3K,L*). However, when a degron allele of Uba2 was degraded, a reproducible delay in MI was observed and spore viability was reduced to 74% compared to 93% in the no auxin control (*Figure 3—figure supplement 2I*). This may be a consequence of more acute inactivation of SUMOylation due to faster and more complete degradation, and/or the fact that Uba2 is the catalytic subunit of E1. Although divisions occurred efficiently when Aos1-AID/Uba2-AID were degraded, JM resolution and crossover formation were delayed by ~30 min, and final crossover levels were reduced by ~18% (*Figure 3K,L* and *Figure 3—figure supplement 2H,I*). These HR defects were accompanied by a delay in de-synapsis (*Figure 3M* and *Figure 3—figure supplement 2J*). Without auxin, only 25% of cells still had partial or full synapsis 1 hr after *NDT80-IN* expression, compared to 57% when Aos1-AID was degraded. After 2 hr, SCs had completely disassembled in 82% of control cells without auxin, compared to 47% following Aos1-AID degradation. Late defects caused by E1 inactivation could reflect SUMO targets such as Sgs1, Top3, Slx4, Smc5/6, Chd1, the ZMM proteins and components of SCs (discussed below, Figures 6 and 7).

Collectively, real-time inactivation of the SUMO E1 enzyme confirms that SUMOylation regulates the major transitions of meiotic prophase and identifies roles in S-phase, DSB formation, the formation and resolution of joint molecules, synapsis and de-synapsis, and the progression of meiotic prophase I.

## Functional classes of meiotic SUMO targets

Proteins involved in all aspect of meiotic chromosome metabolism were SUMOylated, including DNA replication and repair, the DNA damage response, chromatin, transcription, telomeres, homologous recombination, synapsis, and chromosome segregation. Below, we analyze subsets of these targets pertinent for regulation the SUMO and ubiquitin modification systems, chromatin, chromosome structure, and homologous recombination.

## The SUMO machinery
### SUMO (Smt3)

A principal target of meiotic SUMOylation was the SUMO machinery (*Figure 4A*). SUMO itself was the fifth largest contributor to total LFQ signal intensity and all nine lysines on Smt3 were identified as acceptor sites (*Figure 4B*). K15 was the dominant linkage followed by K11 and K19, consistent with SUMO chains being linked primarily via the flexible N-terminal extension (*Klug et al., 2013*).

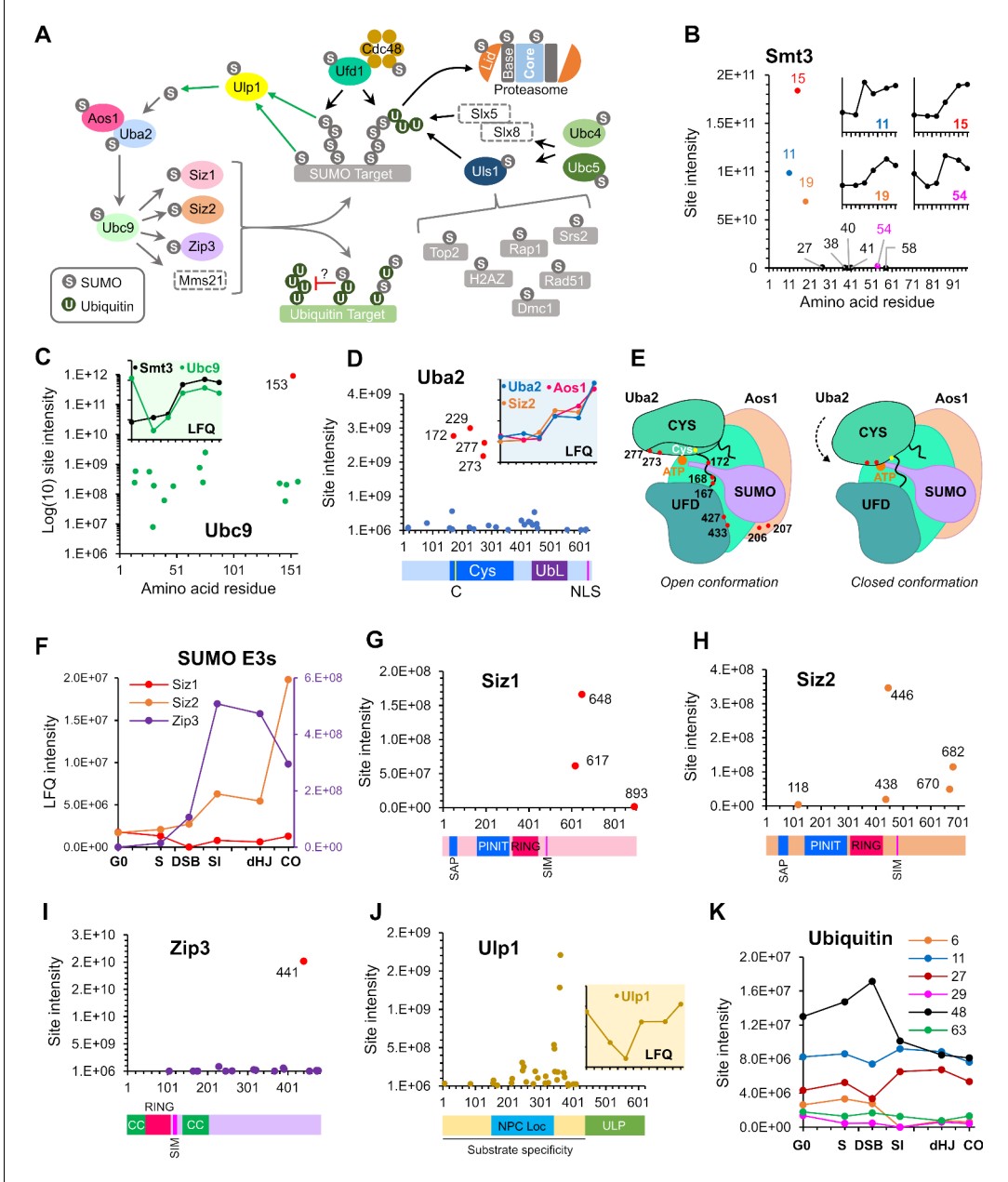

**Figure 4.** SUMOylation of the SUMO and Ubiquitin-proteasome machinery. (**A**) Summary of targets in the SUMO and ubiquitin-proteasome systems and the relationships between these factors. Dashed rectangles indicate SUMOylation was not detected. Note that although SUMOylation of Mms21 was not detected, the associated Smc5/6 complex was modified. (**B**) Intensity plot of SUMOylated sites on Smt3. Insets show temporal profiles of the four most prominent SUMO-chain linkages. (**C**) Intensity plot of SUMOylated sites on E2 conjugase Ubc9. Inset shows normalized LFQ profiles of Ubc9 and Smt3. (**D**) SUMO-site intensities on Uba2. The cartoon below shows key domains along the peptide backbone. C: active-site cysteine, NLS: nuclear localization sequence. Inset shows normalized LFQ profiles of Uba2, Aos1, and Siz2 (**E**) Cartoon of the Aos1-Uba2 structure highlighting the locations of SUMOylation sites. (**F**) LFQ profiles of the SUMO E3 ligases. The Zip3 profile is plotted on the purple Y-axis on the right-hand side. (**G**) (**H**) (**I**) Site intensities of Siz1 (**G**), Siz2 (**H**) and Zip3 (**I**) plotted over their respective domain structures. SAP, SAF-A/B, Acinus, and PIAS domain; PINIT, PINIT motif-containing domain; RING: SpRING domain; SIM, SUMO interacting motif; CC, coiled coil. (**J**) SUMO-site intensities of Ulp1 plotted over its domain structure. NPC Loc, nuclear pore complex localization domain; ULP, Ubiquitin-like protease domain. Inset shows the normalized LFQ profile. (**K**) Intensity profiles of SUMOylation sites on Ubiquitin.

The online version of this article includes the following figure supplement(s) for figure 4:

**Figure supplement 1.** Diverse temporal profiles of SUMOylation sites on the SUMO E1 subunit Uba2.

SUMO chains are essential for meiosis and have been implicated in SC formation and inter-homolog recombination (*Cheng et al., 2006*; *Lin et al., 2010*). Consistently, SUMO chains increased sharply after DSB formation and peaked in pachytene (*Figure 4B* and inset in *4C*). The next most abundant linkage, K54, had the same profile, suggesting a possible role for K54-linked chains in meiotic pro-phase. The most prominent meiotic substrate of poly-SUMOylation is the SC central element protein Ecm11 (*Leung et al., 2015*) and the LFQ profiles of SUMO chains and Ecm11-SUMO were closely matched (*Figure 4L*).

## Ubc9 (E2)

Ubc9 was the largest single contributor to total LFQ signal intensity. 15/17 lysines in Ubc9 were identified as SUMO acceptor sites (*Figure 4C*). K153 accounted for the vast majority of the signal with a 365-fold higher intensity than the next most abundant site. Klug et al. showed that K153-SUMO inhibits conjugase activity and converts Ubc9 into a cofactor that enhances chain formation by unmodified Ubc9 (*Klug et al., 2013*). This activity is particularly important during meiosis where it facilitates SC formation. Ubc9-K153-SUMO was high in G0, dropped in S phase and then increased again at the time of SC formation. High levels of Ubc9-K153-SUMO in G0 seem paradoxical because SUMO chains were relatively low at this time (*Figure 4C*, inset). These observations suggest that Ubc9-K153-SUMO is not the sole regulator of SUMO chain formation. Alternatively, or in addition, SUMO chains may be less stable during G0.

Whether SUMOylation at other sites of Ubc9 plays a function in meiosis is unknown. Knipscheer et al. showed that SUMOylation of human Ubc9 at K14 regulates target discrimination (*Knipscheer et al., 2008*). K76 SUMOylation also has the potential to alter target discrimination as this residue lies in a basic patch that mediates Ubc9 specificity for acidic and phosphorylation-dependent consensus conjugation sites (*Mohideen et al., 2009*). Intriguingly, K76 SUMOylation was highest at G0, but diminished by the time of SC formation after which K153 completely dominated.

## Aos1$^{SAE1}$-Uba2$^{SAE2}$ (E1)

Twenty-seven acceptor sites (57% of all lysines) were identified on Uba2$^{SAE2}$, the catalytic subunit of the E1 activating enzyme, Aos1$^{SAE1}$-Uba2$^{SAE2}$ (*Figure 4D*). LFQ identified four prominent sites, K172, 229, 273, and 277, with (intensities ~ 4-fold higher than the next most intense site). SUMOyla-tion of K172 (or adjacent K167 and K168) is expected to be inhibitory because it lies in the crossover loop involved in the conformational changes that accompany activation (*Olsen et al., 2010*; *Figure 4E*). Although a triple (167,168,172) K-R mutant was previously shown to have negligi-ble impact on the growth and genotoxin sensitivity of mitotically cycling cells (*Albuquerque et al., 2015*), its effects on meiosis are unknown. K273 and K277 lie proximal to the adenylate in the closed conformation of the E1 and are also predicted to be inhibitory. SUMOylation at both sites rises after G0 and then remains relatively constant until the final crossover time point, when levels spike sharply. Also notable is a cluster of eight acceptor sites that mapped around the hinge of the ubiqui-tin-fold domain (residues 400–460) and could also influence conformational changes involved in acti-vation. The diverse intensity-profiles of individual sites suggest that Uba2 regulation may be complex (*Figure 4—figure supplement 1*). Finally, in Aos1$^{SAE1}$, a cluster of four acceptor sites, cen-tered on the intense K207 site, lies close to an insertion loop that could influence adenylation and transfer of SUMO to Ubc9. Aos1-K207 SUMOylation rose steadily after DSB formation.

## E3 ligases, Siz1, Siz2, and Zip3

SUMOylation of an E3 can enhance apparent catalytic activity by providing SUMO in cis to interact with the 'backside' of Ubc9 and thereby stimulate its activity (*Cappadocia et al., 2015*). SUMO con-jugation could also modulate E3 activity in other ways. Three of the four known yeast SUMO E3s, Siz1, Siz2, and Zip3, were modified in meiosis and showed diverse SUMOylation profiles (*Figure 4F*; modification of Mms21 was not detected). Siz1 SUMOylation on three C-terminal tail sites, domi-nated by K648 (*Figure 4G*), was relatively high in G0 and S-phase, dipped during DSB formation and returned thereafter. This profile may reflect known substrates of Siz1 such as PCNA, the septin Cdc3, and transcriptional repressors Sum1 and Tup1, all of which were SUMOylated during meiosis (*Supplementary file 4*; *Johnson and Gupta, 2001*; *Papouli et al., 2005*; *Pfander et al., 2005*;

*Srikumar et al., 2013b*). Mid-prophase targets of Siz1 also include the SC component Ecm11 (*Leung et al., 2015*).

Ecm11 can also be targeted by Siz2, which was SUMOylated at five sites. While two acceptor sites reside in the C-terminal tail of Siz2, two other sites lie between the SP-CTD and the SIM, including the most intense site K446 (*Figure 4H*). Modification at K438 or K446 could provide a well-positioned 'backside SUMO$^B$' in cis to stimulate Siz2 activity (*Streich and Lima, 2016*). Distinct from Siz1, SUMOylation of Siz2 was relatively low during S phase, but rose after DSB formation and showed a large increase during CO formation (*Figure 4F*), closely matching the profiles of Aos1 and Uba2 (*Figure 4D*, inset), and the silencing factor, Esc1 (Figure 7N).

Zip3 is a meiosis-specific SUMO ligase that promotes the chromosomal localization of the ZMM group of pro-crossover factors, and helps locally couple SC formation to prospective CO sites (*Cheng et al., 2006*; *Agarwal and Roeder, 2000*; *Chua and Roeder, 1998*; *Shinohara et al., 2008*). Zip3 also helps recruit Ubc9 and 26S proteasomes to meiotic chromosomes, reminiscent of the SUMO-Ubiquitin axis described in mammals (*Ahuja et al., 2017*; *Hooker and Roeder, 2006*; *Rao et al., 2017*). Zip3 comprises an N-terminal RING domain followed by an essential SIM, a short region of putative coiled-coil and a disordered serine-rich tail. The 17 acceptor sites mapped in Zip3 were scattered throughout the coiled-coil and tail (*Figure 4I*). The one exception, K105 lies adjacent to the SIM, with potential for auto regulation. Sites within the coiled-coil (K165, 175, 176) could influence Zip3 oligomerization and partner interactions. However, SUMOylation at K441 accounted for almost 90% of the total intensity. Intriguingly, this site appears to be part of a compound modification motif, 441-KRSNSTQ-447, comprising a consensus site for the DNA damage-response kinases Mec1/Tel1 (S/T-Q), phosphorylation of which could prime Cdc7-mediated phosphorylation of upstream serines to create an acidic consensus site for SUMOylation, K-x-S(P). Consistent with this idea, phosphorylation of these sites was identified in our dataset, with co-modification being detected on ~2/3rds of the peptides (*Supplementary file 2*). Serrentino et al. previously showed that the four S/T-Q sites of Zip3 are required for its efficient accumulation at recombination sites and full crossover function (*Serrentino et al., 2013*). Zip3-K441-SUMO is possibly the downstream effector of Mec1/Tel1 phosphorylation.

## SUMO isopeptidase Ulp1

The essential Ulp1 isopeptidase matures the SUMO pro-peptide and de-SUMOylates a subset of substrates (*Li and Hochstrasser, 1999*; *Zhao, 2018*). In mitotically cycling cells, Ulp1 regulates myriad processes including DSB repair, 2 μm copy number, mRNA surveillance, silencing, and nuclear import and export of pre-60S ribosomes, but its roles in meiosis have not been characterized (*Dobson et al., 2005*; *Lewis et al., 2007*; *Palancade et al., 2007*; *Panse et al., 2006*; *Stade et al., 2002*; *Zhao et al., 2004*). A remarkable 32 acceptor-sites were mapped across the N-terminal region of Ulp1, but were excluded from the essential C-terminal catalytic domain (*Figure 4J*; *Li and Hochstrasser, 2003*; *Mossessova and Lima, 2000*; *Panse et al., 2003*). Most sites locate in overlapping regions required for tethering Ulp1 to nuclear pores (144–346) and for substrate specificity (amino acids 1–417; *Li and Hochstrasser, 2003*; *Panse et al., 2003*), pointing to roles for SUMOylation in these processes. The overall signal for Ulp1-SUMO dipped around the time of DSB formation but increased again with synapsis, suggesting global changes in protein de-SUMOylation at these times (*Figure 4J*, inset).

## Cross-talk with ubiquitin and autophagy

SUMO targets also revealed cross-talk with the ubiquitin-proteasome system (UPS; *Figure 4A*). UPS targets included Ubc4 and Ubc5, paralogous E2 conjugases involved in protein quality control, stress response, and cell-cycle regulation by the APC/C (*Finley et al., 2012*). Notably, Ubc4/5 work with SUMO-targeted ubiquitin ligases (STUbLs) that target poly-SUMOylated substrates (*Sriramachandran and Dohmen, 2014*; *Uzunova et al., 2007*). SUMO also modified five ubiquitin E3 ligases, Ubr1, Ufo1, Uls1, Gid7, and Ufd4, involved in a variety processes including the N-end rule, catabolite-degradation of fructose-1,6-bisphosphatase, transcription, cell cycle, and genome maintenance (*Bao et al., 2015*; *Baranes-Bachar et al., 2008*; *Finley et al., 2012*; *Kramarz et al., 2017*; *Lin et al., 2015*; *Sriramachandran and Dohmen, 2014*; *Varshavsky, 1997*). Notable is Uls1, a putative STUbL that is thought to compete with a second STUbL, Slx5-Slx8, to displace from DNA

poly-SUMOylated proteins that have been rendered defective or inactive (*Sriramachandran and Dohmen, 2014*; *Tan et al., 2013*; *Wei et al., 2017*). Inferred Uls1 targets, all of which were SUMOylated in meiosis, include Top2 (*Wei et al., 2017*), H2A.Z (*Takahashi et al., 2017*), Rap1 (*Lescasse et al., 2013*), Srs2 (*Kramarz et al., 2017*), Rad51 (*Chi et al., 2011*) and possibly Dmc1 (*Dresser et al., 1997*; *Figure 4A*). The ubiquitin-dependent segregase, Cdc48/p97, can target proteins co-modified by ubiquitin and SUMO (*Bergink et al., 2013*; *Nie et al., 2012*). Both Cdc48 and Ufd1, the cofactor implicated in SUMO binding, were SUMOylated (*Figure 4A*).

Intriguingly, ubiquitin itself was SUMOylated on six lysines that are also sites for ubiquitin chain formation (K6, 11, 27, 29, 48, and 63; *Figure 4K*). Clear differences in site intensities were detected, with K48 and K11 being the most abundant followed by K27. Interestingly, K48 SUMOylation, which has the potential to modulate targeting to proteasomes, peaked during DSB formation and dipped thereafter. Ubiquitin is expressed from four loci in budding yeast, as a head-to-tail poly-ubiquitin precursor from the *UBI4* locus, and as ubiquitin fusions to the ribosomal proteins Rps31 and Rpl40A/Rpl40B. Whether ubiquitin SUMOylation reflects modification of ubiquitin precursors, free ubiquitin and/or ubiquitin conjugates is unclear. However, this observation further corroborates evidence for a unique class of mixed ubiquitin-SUMO chains with potential for novel signaling functions (*Esteras et al., 2017*; *Hendriks et al., 2014*).

Autophagy is particularly important for the initiation of meiosis (*Wen et al., 2016*), but also functions in nuclear architecture and chromosome segregation (*Matsuhara and Yamamoto, 2016*). Three autophagy factors were SUMOylated: the ubiquitin-like Atg8 protein that undergoes lipidation via conjugation to phosphatidylethanolamine, its cognate E1 enzyme Atg7, and the dual receptor Cue5, which simultaneously binds ubiquitylated cargo and Atg8 (*Supplementary file 4*). These conjugates raise the possibility that SUMO modulates the targeting of aggregated proteins and inactivated proteasomes to phagophores (*Wen and Klionsky, 2016*).

## SMCs, homolog axes, and synaptonemal complex

### SMC (structural maintenance of chromosomes) complexes

SMC complexes are fundamental to meiotic chromosome metabolism, mediating sister-chromatid cohesion, the organization of chromosomes into linear arrays of chromatin loops, chromosome compaction and segregation, and regulating all aspects of recombination (*Hunter, 2015*; *Jessberger, 2002*; *Zickler and Kleckner, 2015*). Mapping meiotic SUMOylation sites onto the six SMC proteins revealed a strong bias for modification of the coiled-coil regions (*Figure 5A*, *Figure 5—figure supplements 1–3*). Helical projections showed that the majority of the sites are solvent exposed and therefore unlikely to alter underlying coiled-coil structures (*Figure 5—figure supplements 1–2*). Given this general propensity and the similar SUMOylation profiles (*Figure 5B*), it is possible that SUMOylation of all SMC proteins has a common function. For example, SUMO could compete or synergize with acetylation to regulate interactions between SMC coiled coils (*Kulemzina et al., 2016*); or could influence folding at the elbow region (*Bürmann et al., 2019*).

### Cohesin

All components of meiotic cohesin were SUMOylated on multiple sites: Smc1 (16 sites), Smc3 (14), Pds5 (6), Irr1/Scc3 (3), and the meiosis-specific kleisin Rec8 (16). Even though cohesin components other than Rec8 are not meiosis specific, SUMOylation was largely undetectable in G0 (*Figure 5C*). Rec8-SUMO increased rapidly by S-phase, continued to increase through DSB and SI time points and remained relatively high thereafter. SUMOylations of Smc1 and Smc3 were less intense than Rec8 and although low levels were detected in S phase, big increases were seen later, after DSB formation (*Figure 5B and C*). These profiles are consonant with studies in mitotically cycling cells indicating that SUMOylation is essential for establishment of cohesion in S phase (*Almedawar et al., 2012*), but also point to later roles. For example, Rec8 SUMOylation could influence its interactions with the HEAT repeat proteins, Pds5 and Irr1/Scc3.

Intriguingly, Pds5 somehow protects the mitotic Kleisin, Mcd1/Scc1 from poly-SUMOylation and ensuing Slx5/8-dependent degradation (*D'Ambrosio and Lavoie, 2014*) consistent with the observation that overexpression of the Ulp2 SUMO isopeptidase suppresses *pds5* mutant phenotypes (*Stead et al., 2003*). However, SUMOylation of Mcd1/Scc1 is also important for damage-induced cohesion in mitotic cells (*McAleenan et al., 2012*; *Wu et al., 2012*). Pds5 and Rec8 may share similar

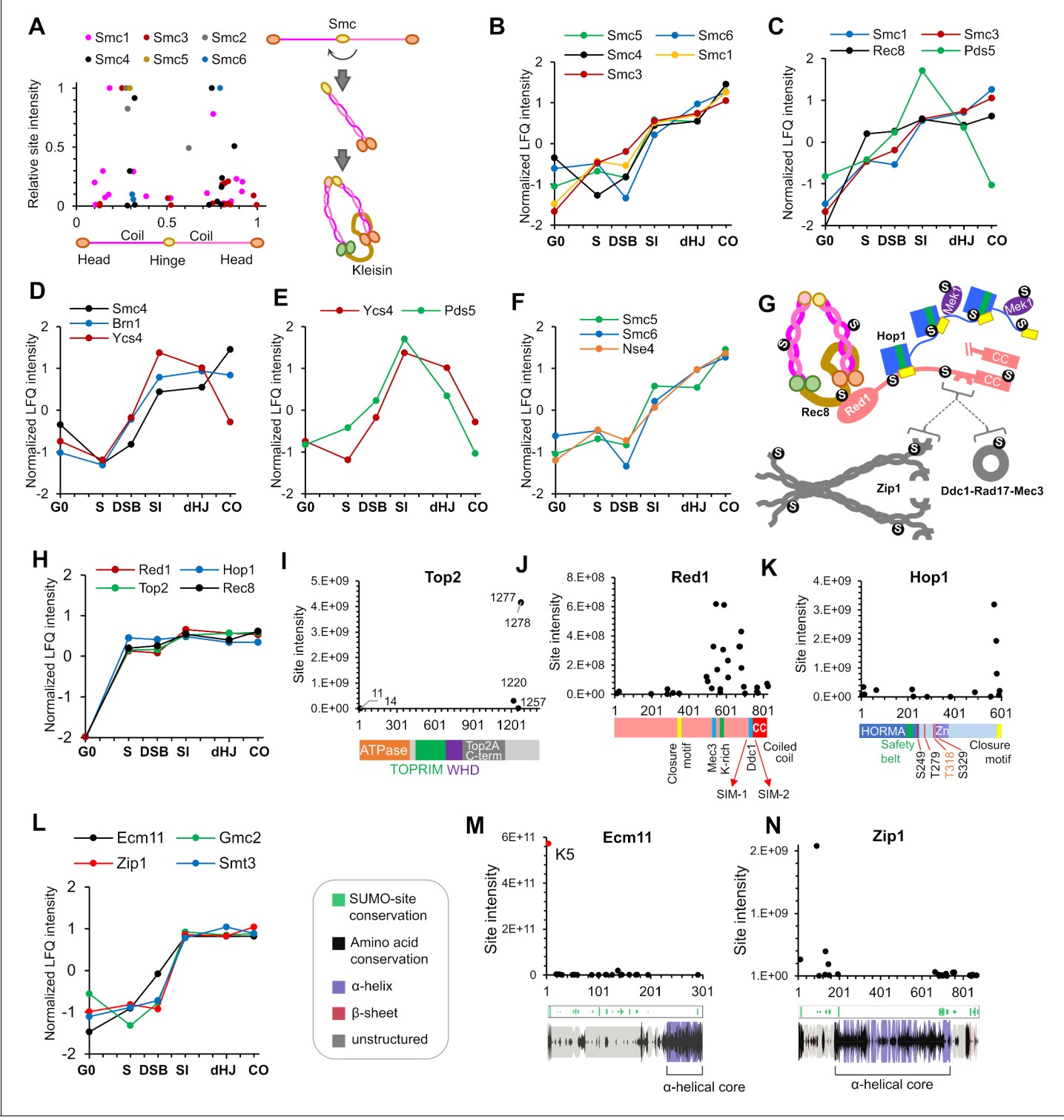

**Figure 5.** SUMOylation of homolog axes and synaptonemal complex. (**A**) Relative positions of SUMO sites on the SMC proteins. Adjacent cartoon illustrates how SMCs are assembled into ring structures. (**B–F**) Normalized LFQ profiles of (**B**) the SMCs (note that Smc2 did not have measurable LFQ intensities); (**C**) components of Rec8-cohesin; (**D**) condensin; (**E**) HEAT-repeat proteins Ycs4 and Pds5; and (**F**) components of the Smc5/6 complex. (**G**) Summary of SUMOylated axis proteins and pertinent interaction partners. Arcs in Red1 and Zip1 indicate SIMs. (**H**) Normalized LFQ profiles of axis components Red1, Hop1, Top2, and Rec8. (**I**) Site intensities of Top2 plotted along its domain structure. TOPRIM, Topoisomerase-primase domain; WHD, winged-helix domain. (**J**) Site intensities of Red1 plotted along its domain structure. (**K**) Site intensities of Hop1 plotted along its domain structure. S249, T279 T318, and S329 are key phosphorylation sites important for Hop1 function. (**L**) Normalized LFQ profiles of SC central region

*Figure 5 continued on next page*

*Figure 5 continued*

components Ecm11, Zip1, and Gmc2 relative to SUMO (Smt3). (M) (N) Site intensity plots for (M) Ecm11 and (N) Zip1 plotted along predicted secondary structures.

The online version of this article includes the following figure supplement(s) for figure 5:

**Figure supplement 1.** Secondary structure and helical projections of Cohesin SMC proteins.
**Figure supplement 2.** Secondary structure and helical projections of Condensin SMC proteins.
**Figure supplement 3.** Secondary structure and helical projections of Smc5.

relationships during meiosis. Rising SUMOylation of Smc1/3 at the SI timepoint was accompanied by transient peaks of Pds5 and Irr1/Scc3 modification (*Figure 5C*; and not shown), which declined thereafter. Pds5-SUMO may be associated with cohesin dynamics occurring as HR ensues and axes mature. The six acceptor sites mapped on Pds5 were dominated by K1103, which is predicted to lie close to the interface with Rec8 and Smc3, with potential to influence ring opening (*Hons et al., 2016*; *Ouyang and Yu, 2017*). Finally, the cohesin loader complex, Scc2-Scc4, had a single low intensity site in Scc2, K76, which maps to the interaction interface.

## Condensin

SUMOylation of all condensin subunits was detected in meiosis and, like cohesin, the Kleisin subunit, Brn1, was the most intense (*Figure 5D* and *Supplementary file 4*). 10 sites were mapped in Brn1 but a single site predominated, K628, which has potential to influence interaction with Ycg1 (*Kschonsak et al., 2017*). Four other sites (K421, 439, 445 and 457) clustered around the safety-belt loop predicted to trap DNA (*Kschonsak et al., 2017*). Unlike cohesin, significant SUMOylation of condensin subunits was detected in G0 and then increased steeply after DSB formation, consistent with roles after meiotic S-phase (*Yu and Koshland, 2003*). The modification profile of Brn1 is similar to proteins involved in SC formation (Zip1, Zip3, Smt3, Ubc9, Ecm11, and Gmc2); and unlike that of Rec8, which shows high levels of SUMOylation already in S-phase and closely matches the profiles of other axis proteins (Hop1, Red1, and Top2, see below and *Figure 5G*). These observations are consistent with later loading and function of condensin in meiotic prophase. During mitosis, condensin is targeted to the rDNA in part via Ycs4 SUMOylation, which is promoted by the FEAR kinase, Cdc14 (*D'Amours et al., 2004*). In meiotic prophase, Ycs4-SUMO peaked at the SI time point with a profile analogous to that of Pds5, perhaps reflecting a similar function in regulating condensin dynamics in response to DSBs and nascent synapsis (*Figure 5D and E*).

## Smc5/6 complex

The third eukaryotic SMC complex facilitates DNA replication, repair and recombination and interacts with a dedicated SUMO E3 ligase, Nse2/Mms21, via the Smc5 subunit, and with the Siz1/2 ligases via Nse5 (*Bustard et al., 2016*; *Uhlmann, 2016*). In meiosis, Smc5/6 facilitates JM formation and resolution likely through regulation of the Mus81-Mms4 and Sgs1-Top3-Rmi1 complexes (*Copsey et al., 2013*; *Lilienthal et al., 2013*; *Xaver et al., 2013*). SUMOylation of Smc5, Smc6 and Kleisin Nse4, but not the other five subunits of the complex, was detected in meiosis. Like cohesin, the Smc5/6 complex was not SUMOylated in G0, increased in S phase, as expected, dipped as DSBs formed and then increased thereafter consistent with its roles throughout meiotic prophase (*Figure 5F*; *Copsey et al., 2013*; *Lilienthal et al., 2013*; *Xaver et al., 2013*).

## Axis components Hop1, Red1, and Top2

Red1 and Hop1 augment cohesin-based homolog axes and play essential roles in HR, synapsis and checkpoint control (*Hunter, 2006*; *Subramanian and Hochwagen, 2014*; *Zickler and Kleckner, 2015*). Red1 appears to be recruited to axes principally via an interaction with Rec8, while Hop1 interacts with Red1 (*Figure 5G*; *Smith and Roeder, 1997*; *Sun et al., 2015*; *West et al., 2019*; *West et al., 2018*; *Woltering et al., 2000*). Type-II topoisomerase Top2 is an additional axial component implicated in loop-axis organization and crossover interference (*Klein et al., 1992*; *Moens and Earnshaw, 1989*; *Zhang et al., 2014*). The physical and functional interactions between Rec8, Red1, Hop1, and Top2 were reflected in their SUMOylation profiles, which emerged as a tightly clustered cohort (*Figure 5H*).

In somatic cells, SUMOylation of Top2 has been implicated in multiple aspects of centromere function as well as the removal of abortive cleavage complexes (*Cubeñas-Potts and Matunis, 2013*; *Wei et al., 2018*; *Zhao, 2018*). In meiosis, Top2-SUMO facilitates crossover interference together with Red1-SUMO, Sir2 and the STUbL Slx5-Slx8 (*Zhang et al., 2014*). Our analysis identified two relatively weak N-terminal SUMO sites in addition to the previously mapped C-terminal sites (*Figure 5I*). In Red1, a remarkable 32 conjugated sites were identified, 20 of which clustered into a broad domain (492-702) surrounding a lysine-rich patch previously shown to be important for Red1 SUMOylation, SC assembly, and crossover interference (*Figure 5J* and *Supplementary file 4*; *Eichinger and Jentsch, 2010*; *Zhang et al., 2014*). These include sites that overlap regions shown to interact with Mec3 and Ddc1, components of the 9-1-1 checkpoint-sensor clamp (*Eichinger and Jentsch, 2010*; *Zhang et al., 2014*), as well as two SUMO-interaction motifs inferred to mediate interaction with the SC transverse-filament protein, Zip1 (*Lin et al., 2010*). Additional sites were located at the Red1 closure motif, which is bound by Hop1; and in the C-terminal coiled-coil motif that mediates Red1 oligomerization (*West et al., 2019*; *West et al., 2018*). Thus, SUMO may influence all aspects of Red1 function.

Hop1 self-oligomerizes via interactions between the N-terminal HORMA domain and a C-terminal closure motif and thereby activates the checkpoint kinase Mek1 (*Figure 5G and K*; *West et al., 2018*). The HORMA domain also anchors Hop1 to homolog axes by interacting with a closure motif in Red1. 17 SUMO-conjugation sites were scattered throughout Hop1, including an N-terminal cluster and sites located around the 'safety belt', which wraps the closure motifs. However, the highest-intensity sites were found immediately adjacent to the closure motif, suggesting effects on oligomerization (*Figure 5K*).

## Synaptonemal complex components Zip1, Ecm11, and Gmc2

SUMO has well documented roles in SC assembly: it localizes to the SC central region and SUMO chains are required for SC formation (*Cheng et al., 2006*; *Voelkel-Meiman et al., 2013*). Consistently, SUMO chain formation was concurrent with the SUMOylation of SC central region proteins, Zip1, Ecm11, and Gmc2 (*Figure 5L*). Zip1 activates SUMOylation of Ecm11, which then stimulates Zip1 to oligomerize (*Leung et al., 2015*). 20 of the 21 conjugation sites in Ecm11 mapped throughout the predicted unstructured N-terminal region, but SUMOylation at K5 was over four orders of magnitude more intense than at any other site (*Figure 5M* and *Supplementary file 4*). K5 was previously shown to be responsible for most Ecm11 SUMOylation and is essential for its function, underscoring the value of our LFQ analysis for identifying functionally relevant conjugation sites (*Humphryes et al., 2013*; *Zavec et al., 2008*). Gmc2, the partner of Ecm11, was also SUMOylated at six sites (*Supplementary file 4*).

The 23 sites mapped in Zip1 included two clusters at the boundaries between the central helical core and the disordered N- and C-terminal regions (*Figure 5N*). In the context of the bifurcated tetramer model of *Dunce et al., 2018*, SUMO site locations point to possible roles in self-assembly of the Zip1 zipper-like lattice, both at the αN-end head-to-head assembly and the αC-end tetramization regions. Consistent with this possibility, residues 21–163, encompassing the most intense Zip1 SUMO sites, are important for SC assembly and Ecm11 SUMOylation (*Voelkel-Meiman et al., 2016*). The most N-terminal site, K13, lies in a region important for crossing over and recruitment of the E3 ligase, Zip3 (*Voelkel-Meiman et al., 2019*). A C-terminal cluster of SUMO sites (K833, 841, 847, 854, and 867) overlaps with a region of Zip1 that interacts with Red1 and includes a SIM that is required for Zip1-Red1 interaction (Zip1 residues 853–864, *Figure 5G and N*; *Lin et al., 2010*). Thus, SUMOylation of Zip1 may modulate its oligomerization, axis association, and interaction with Zip3 and perhaps other HR factors.

## Homologous recombination

Meiotic HR is physically and functionally linked to homolog axes and synaptonemal complexes (summarized in *Figure 6A*; *Zickler and Kleckner, 2015*), and our functional analysis implies that each step of meiotic HR is regulated by SUMO (*Figure 3*).

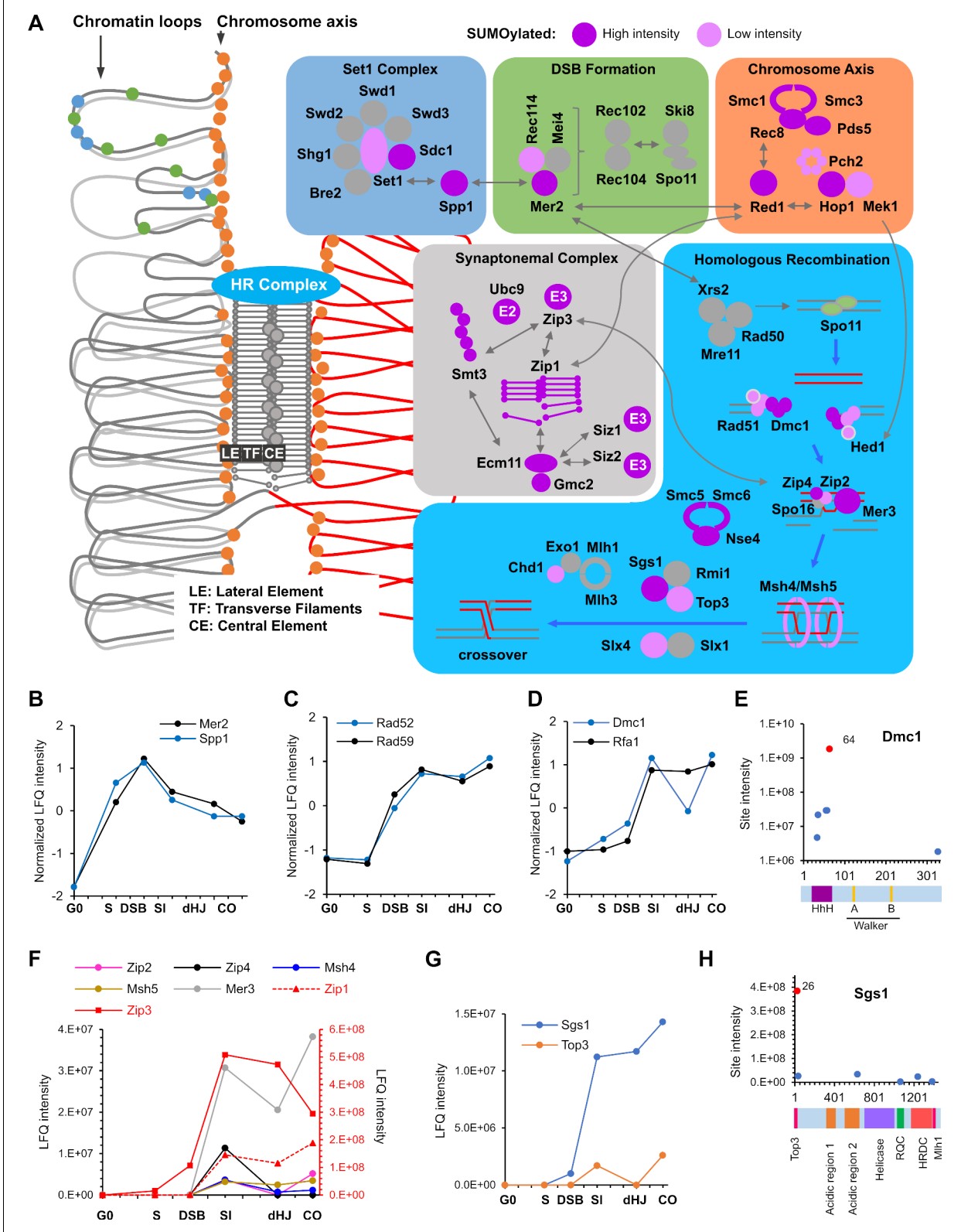

**Figure 6.** SUMOylation of the recombination machinery. (A) Summary of SUMOylated meiotic HR factors and their functional integration with other pertinent SUMO targets in the chromosome axes and SCs. The schematic on the left-hand side illustrates the linear chromatin loop array of prophase-I chromosomes and relative localizations of SUMOylated factors illustrated in the color-coded boxes. (B–D) Normalized LFQ profiles of: (B) DSB factors Mer2 and Spp1; (C) Rad51-mediator and DNA strand-annealing proteins Rad52 and Rad59; and (D) the strand exchange factors Dmc1 and Rfa1. (E) Site

*Figure 6 continued on next page*

*Figure 6 continued*

intensities of Dmc1 plotted along its domain structure. HhH: Helix-hairpin-helix. (F) LFQ profiles of the ZMM proteins (SUMOylation of Spo16 was not detected). Zip1 and Zip3 are plotted on the right-hand side Y-axis in red. (G) Normalized LFQ profiles of DNA helicase Sgs1 and topoisomerase Top3. (H) Sgs1 site intensities mapped along its domain structure. Top3, Top3 interacting domain; RQC, RecQ C-terminal domain; HRDC, helicase and RNaseD C-terminal domain; Mlh1, Mlh1 interacting domain.

## DSB formation and processing

Spo11-catalyzed DSBs occur in narrow hotspots located in gene promoters marked by H3K4 trimethylation (*Pan et al., 2011*). Hotspots map to chromatin loops, while the DSB machinery localizes to the axes (*Blat et al., 2002*; *Pan et al., 2011*; *Panizza et al., 2011*). Through the formation of tethered loop-axis complexes, loop sequences become axis associated and DSB formation is triggered (*Blat et al., 2002*; *Lam and Keeney, 2015*). 10 factors that assemble into four complexes are essential for DSB formation, *viz.* Spo11-Ski8, Rec102-Rec104, Mer2$^{IHO1}$-Rec114-Mei4, and Mre11-Rad50-Xrs2$^{NBS1}$ (MRX) (*Lam and Keeney, 2015*). DSB formation is also strongly, although not completely, dependent on axis associated factors Hop1, Red1, and Rec8-cohesin, all of which were heavily SUMOylated, as described above (*Figure 5*). Intriguingly, two key regulatory DSB factors, Spp1 and Mer2, were SUMOylated at multiple sites (*Figure 6B*). Spp1 reads H3K4 trimethylation marks using its PHD finger and recruits DSB hotspots to the chromosome axis via an interaction with Mer2 (*Acquaviva et al., 2013*; *Adam et al., 2018*; *Sommermeyer et al., 2013*). Mer2$^{IHO1}$ is recruited to axes via Red1 and Hop1, where its cell cycle and replication-dependent phosphorylation leads to association with other DSB factors, Rec114-Mei4 and Xrs2, to trigger DSB formation only on replicated sister chromatids (*Lam and Keeney, 2015*). Our data revealed sharp SUMOylation peaks for both Spp1 and Mer2 at the DSB time point (*Figure 6B*), suggesting roles in initiating HR. The 13 SUMO sites in Mer2 do not overlap with known phosphorylation sites suggesting parallel regulation. Low-intensity SUMO sites were also identified on Rec114 and two subunits of the H3K4 methyltransferase (*Figure 6A* and *Supplementary file 4*).

Processing of DSB ends to liberate Spo11-oligo complexes and produce long 3' single-stranded tails involves Sae2$^{CtIP}$, the endonuclease and 3'−5' exonuclease activities of the MRX complex, and the 5'−3' exonuclease Exo1 (*Lam and Keeney, 2015*; *Zakharyevich et al., 2010*). Only SUMOylation of Sae2$^{CtIP}$ was detected in meiosis, at a site previously shown to enhance Sae2$^{CtIP}$ solubility and facilitate its DSB resection function (*Sarangi et al., 2015*). However, Sae2-SUMO signal was weak and only detected in G0 and S time-point samples.

## Homology search and DNA strand exchange

Mediators help the strand-exchange proteins, Rad51 and Dmc1, displace RPA from single-stranded DNA and thereby assemble into nucleoprotein filaments (*Brown and Bishop, 2015*). Mediators include Rad52, which was SUMOylated at 10 sites, including those previously shown to be important in vegetative cells (K10/11 and 220). During meiosis, Rad52 mediates assembly of Rad51, but not Dmc1, and is important for post-invasion strand-annealing steps during dHJ formation and non-crossover formation (*Gasior et al., 1998*; *Lao et al., 2008*). The SUMOylation profiles of Rad52 and its paralog, Rad59 were tightly matched, rising at the time of DSB formation and continuing to rise during strand invasion, before rising again at the time of crossing over (*Figure 6C*). Diverse effects of Rad52 SUMOylation have been reported in mitotically cycling cells (*Dhingra and Zhao, 2019*), including both enhancing and disfavoring Rad52-Rad51 interaction and HR; and promoting interaction with Rad59 to favor Rad51-indpendent repair such as single-strand annealing (SSA). Analysis of Rad52/Rad59 SUMOylation in the context of meiotic HR may help disentangle these seemingly contradictory functions.

Rfa1, the large subunit of the trimeric RPA complex, was also SUMOylated at five sites, K68, 170, 180, 411, 427 (*Supplementary file 4*). The latter four sites enhance DNA-damage induced checkpoint signaling in vegetative cells by strengthening interaction with the DNA helicase, Sgs1 (*Dhingra et al., 2019*). Rfa1-SUMO levels were low until the SI time point, during which they increased dramatically and then remained high (*Figure 7D*). This profile argues against an early role in Dmc1/Rad51 assembly and suggests that D-loop associated Rfa1 becomes SUMOylated, perhaps

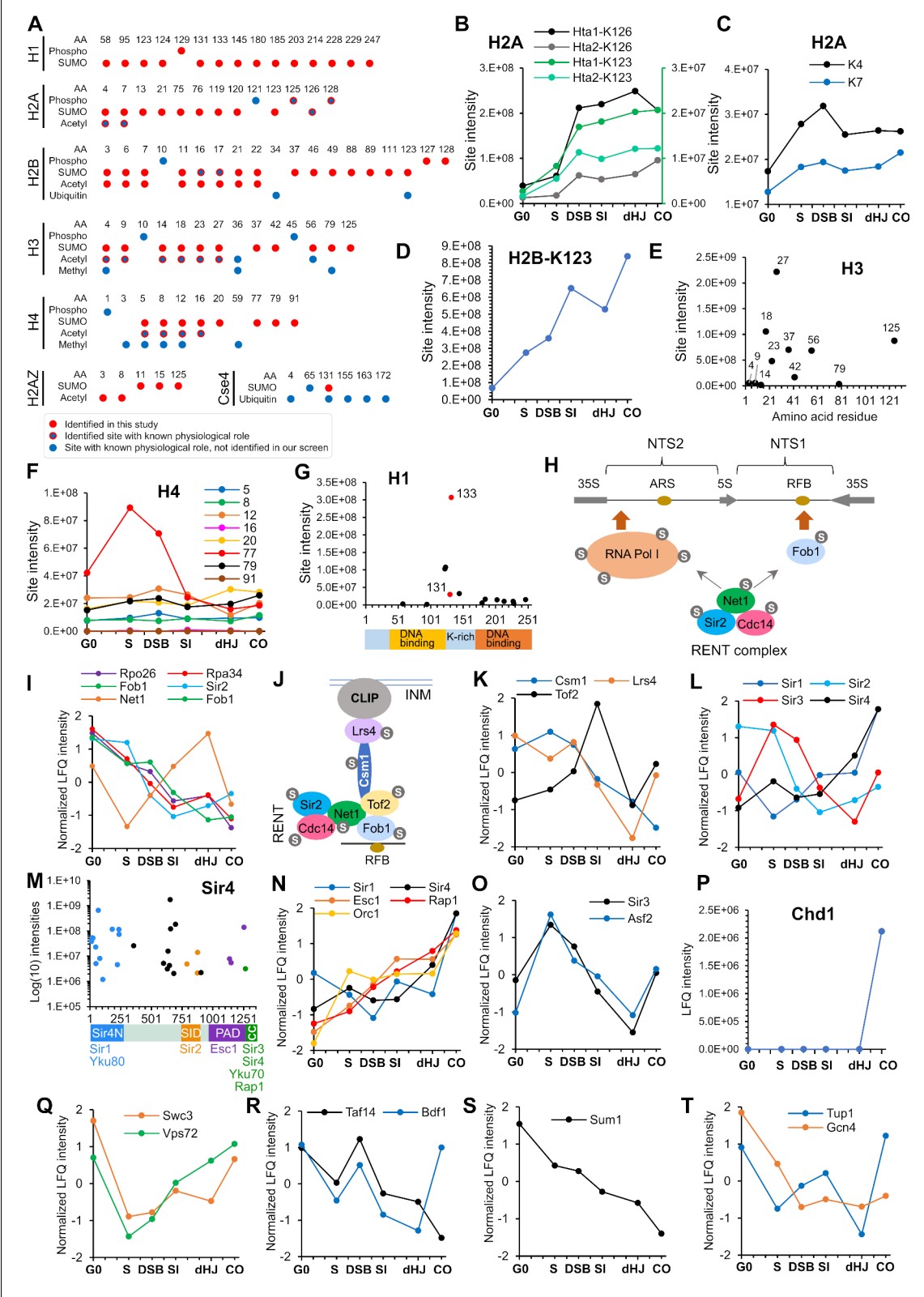

**Figure 7.** SUMOylation of chromatin and associated factors. (**A**) Chart summarizing histone SUMOylation, phosphorylation and acetylation sites identified in this study, together with previously identified modifications. AA, amino acid position. (**B**) Temporal profiles of SUMOylation on K123 and K126 of histones H2A1 and H2A2. The K123 profiles are plotted on the right-hand side Y-axis in green. (**C**) Temporal profiles of K4 and K7 SUMOylation on histone H2A. (**D**) Temporal profile of histone H2B-K123 SUMOylation. (**E**) SUMO site intensities on histone H3. (**F**) Temporal profiles of histone H4

*Figure 7 continued on next page*

*Figure 7 continued*

SUMOylation sites. (**G**) Site intensities of histone H1 SUMOylation sites plotted along its domain structure. Two sites that are co-modified with adjacent phosphorylation are indicated in red. (**H**) Summary of SUMOylated proteins involved in rDNA silencing. NTS1/2, non-transcribed sequences; RFB, replication fork barrier. (**I**) Normalized LFQ profiles of RENT complex components. (**J**) Cartoon illustrating the recruitment of rDNA loci to the inner nuclear membrane (INM) via RENT, cohibin (Csm1 and Lrs4), and CLIP (chromosome linkage inner-nuclear membrane proteins) complexes. INM, inner-nuclear membrane; RFB, replication fork barrier. (**K**) Normalized LFQ profiles of Cohibin (Csm1 and Lrs4) and Tof2. (**L**) Normalized LFQ profiles of the SIR proteins. (**M**) Site intensity plot of Sir4 plotted along its domain structure. (**N**) Normalized LFQ profiles of Sir4 and its binding partners. (**O**) Normalized LFQ profiles of Sir3 and its paralog Asf2. (**P**) LFQ profile of chromatin remodeler Chd1. (**Q**) Normalized LFQ profiles of SWR1-chromatin remodeler components Swc3 and Vps72 (**R**) Normalized LFQ profiles of chromatin remodeler components Bdf1 (SWR subunit that binds acetylated H4) and Taf14 (subunit of TFIID, TFIIF, INO80, Swi/Snf, and NuA3m that binds acetylated H3). (**S**) Normalized LFQ profile of Sum1, a component of the Sum1-Rfm1-Hst1 deacetylase that represses middle-meiotic genes. (**T**) Normalized LFQ profiles of Tup1, a constituent of the Tup1-Cyc8 transcriptional repressor, and the transcriptional activator Gcn4.

to regulate post-invasion steps including DNA synthesis, and strand annealing during dHJ formation and noncrossover formation.

Dmc1 and Rad51 cooperate during meiosis to bias strand-exchange between homologs (*Brown and Bishop, 2015*; *Cloud et al., 2012*; *Lao et al., 2013*). Dmc1 catalyzes the majority of strand-exchange in concert with its essential accessory factor Hop2-Mnd1, while Rad51 plays an essential support function. In fact, efficient inter-homolog bias involves inhibition of Rad51 strand-exchange activity by a meiosis-specific factor Hed1 (*Busygina et al., 2008*). Dmc1, Rad51, Hop2, Mnd1, and Hed1 were all SUMOylated during meiosis (*Supplementary file 4*). Dmc1 was the most prominent of these, with a profile similar to that of Rfa1, except that levels dipped after strand-exchange, at the dHJ time point, before rising again as crossovers formed (*Figure 7D*). Five of the six SUMO sites in Dmc1 clustered in the N-terminal domain (*Figure 7E*), which is conserved in Rad51 and implicated in filament subunit interactions, DNA binding and ATPase activation (*Galkin et al., 2006*; *Zhang et al., 2005*).

## Crossover/noncrossover differentiation

The ZMMs (Zip1, Zip2, Zip3, Zip4, Spo16, Msh4-Msh5, and Mer3) are conserved meiosis-specific pro-crossover factors that locally couple HR to synapsis (*Hunter, 2015*). All ZMMs other than Spo16 were SUMOylated coincident with strand invasion and synapsis (*Figure 6F*). Intensities varied by at least an order of magnitude, with Zip1 and Zip3 (discussed above) being the most prominent (red scale in *Figure 6F*). Both subunits of the HJ-binding MutSγ complex, Msh4-Msh5, were SUMOylated for a total of 11 sites, with the most intense located in the ABC ATPase domains that interface to form two composite ATPase sites (*Rakshambikai et al., 2013*; *Warren et al., 2007*). The DNA helicase Mer3 promotes dHJ formation while limiting the extent of heteroduplex (*Börner et al., 2004*; *Duroc et al., 2017*). Mer3 SUMO sites were located predominantly in the disordered N- and C-termini that may mediate protein-protein interactions and/or protein stability (*Supplementary file 5*).

## Joint molecule (JM) resolution

The conserved endonuclease MutLγ (Mlh1-Mlh3) works with a second DNA mismatch-repair factor Exo1, and the chromatin remodeler Chd1 to promote crossover-biased resolution of dHJs (*Cannavo et al., 2020*; *Kulkarni et al., 2020*; *Wild et al., 2019*; *Zakharyevich et al., 2010*; *Zakharyevich et al., 2012*). As described below, Chd1 was SUMOylated specifically at the time of JM resolution (*Figure 7P*), but modification of MutLγ and Exo1 was not detected. The structure-selective endonuclease Mus81-Mms4^EME1 and the Smc5/6 complex define a second JM resolvase pathway (*Figure 5F*). While modification of Mus81-Mms4^EME1 was not detected, SUMOylation of Smc5/6 components continued to rise during resolution (discussed above and *Figure 5F*). The STR complex, Sgs1^BLM-Top3-Rmi1, is a potent DNA decatenase that functions as a noncrossover-specific dHJ 'dissolvase', but is also a general mediator of JM formation and facilitates MutLγ-Exo1 dependent crossing over (*De Muyt et al., 2012*; *Hunter, 2015*; *Zakharyevich et al., 2012*). In vegetative cells, Sgs1^BLM SUMOylation fosters interaction with Top3, facilitates JM resolution and suppresses crossing over (*Bermúdez-López et al., 2016*; *Bonner et al., 2016*). Both Sgs1^BLM and Top3, but not Rmi1, were SUMOylated during meiosis (*Figure 6G* and *Supplementary file 4*). Sgs1-SUMO rose dramatically with strand exchange and rose again during crossover formation. By comparison, Top3-

SUMO was very weak and only detected in the SI and CO time points. Contrary to the inference that modification of Sgs1 in vegetative cells occurs primarily at three consensus sites (K175, 621, and 831) (*Bermúdez-López et al., 2016*), meiotic SUMOylation occurred at six non- and partial-consensus sites, dominated by K26 (*Figure 6H*). K26 and a second SUMO site, K35, lie in the Top3-interaction domain of Sgs1 and may influence their interaction.

## Chromatin

### Histones

SUMOylation of histones is generally implicated in transcriptional repression by recruiting HDACs, or by competing with activating marks such as acetylation and ubiquitylation (*Cubeñas-Potts and Matunis, 2013*; *Nathan et al., 2006*; *Shiio and Eisenman, 2003*). However, mechanistic details and roles of SUMOylation at individual sites remain largely unexplored. We found that the four core histones, H2A variant Htz1 (H2A.Z), linker histone H1, and the H3-like centromeric histone, Cse4, were SUMOylated in meiosis (*Figure 7A*). In many cases, mapped sites are also known targets of other lysine modifications suggesting antagonistic relationships. However, we also found specific examples of obligate co-modification with phosphorylation or acetylation, pointing to more complex, potentially cooperative interactions between SUMO and other histone marks.

### Histone H2A

Mec1/Tel1-mediated H2A-S128 phosphorylation (γ-H2A) plays important roles in the DNA damage response (*Downs et al., 2000*; *Morrison et al., 2004*; *Redon et al., 2003*; *van Attikum et al., 2004*). In yeast meiosis, γ-H2A is initially triggered in S-phase (*Cheng et al., 2013*) and contributes to a checkpoint that prevents re-replication (*Najor et al., 2016*). In our dataset, Hta1 (H2A1) K123 SUMOylation was only observed on a peptide that was also phosphorylated at either at T125 or S128. Similarly, Hta2 (H2A2) K123 SUMOylation was always coincident with S128 phosphorylation. Although the K123 site does not conform to a canonical phospho-dependent SUMOylation motif (PDSM, ΨKxExxSP), S128 phosphorylation is suitably positioned to enhance its conjugation (*Hietakangas et al., 2006*). Unlike K123, the much more abundant K126 SUMOylation (on either Hta1 or Hta2) was not always coincident with phosphorylation. Thus, SUMOylation of K123, but not K126 appears to be phospho-dependent. Analogous to S128 phosphorylation (*Cheng et al., 2013*), K123 and K126 SUMOylation appeared at the onset of S-phase, increased with DSB formation and then persisted throughout prophase (*Figure 7B*). These observations suggest a compound role for T125/128 phosphorylation and K123 SUMOylation in S-phase and/or recombination.

SUMOylation of H2A at K4 or K7 was always coincident with adjacent acetylation of K7 or K4, respectively (*Figure 7A*), suggesting that N-terminal SUMOylation is stimulated by pre-existing acetylation, as seen for human H3 (*Hendriks et al., 2014*). This subset of H2A modifications showed distinct dynamics relative to those in the C-terminus, with peak intensities in the S-phase and DSB samples (*Figure 7C*), when gene promoters are being targeted for DSB induction. Notably, H2A K5/K8 acetylation leads to Swr1-dependent incorporation of H2A.Z at promoters and activation of transcription (*Altaf et al., 2010*). Possibly, this co-modification event facilitates chromatin changes that modulate DSB formation.

### Histone H2B

N-terminal SUMOylation of H2B (K6, 7, 16, 17) was previously inferred to compete with acetylation to repress transcription (*Nathan et al., 2006*). SUMOylation of these sites was also detected in meiosis, both alone and in combination with acetylation (*Figure 7A*). K123 was the most prominent H2B SUMOylation site, which climbed in intensity throughout prophase I (*Figure 7D*). Ubiquitination at this site appears to have pleiotropic roles in transcriptional activation and elongation, telomeric silencing, DNA replication and DNA repair (*Fuchs and Oren, 2014*). Notably, in meiosis, H2B K123 and the Bre1 ubiquitin ligase are required for efficient DSB formation (*Yamashita et al., 2004*), likely due to the role of H2B-K123 ubiquitylation in stimulating Set1-mediated H3-K4 trimethylation, which in turn helps recruit the DSB machinery to chromatin (*Borde et al., 2009*; *Lam and Keeney, 2015*). Set1 also facilitates progression of meiotic S-phase and the expression of middle meiotic genes (*Sollier et al., 2004*). We suggest that H2B K123 SUMOylation may compete with ubiquitylation to down-regulate H3-K4 methylation and thereby repress middle meiotic genes and attenuate DSB formation in later stages of meiotic prophase.

## Histone H3

Extensive SUMOylation and acetylation of the N-terminus of H3 was also detected (*Figure 7A*). This included H3-K4 suggesting that SUMOylation of a second histone site might also negatively regulate DSB formation. However, SUMOylation at K4, 9 and 14 was much less intense than at the adjacent sites, K18, 23, and 27 (*Figure 7E*).

## Histone H4

SUMOylation of H4 functions in transcriptional repression (*Nathan et al., 2006*; *Shiio and Eisenman, 2003*). We found extensive SUMOylation of the H4 N-terminus including K5, 16, and 20, which was always coincident with acetylation (*Figure 7A and F*). Interestingly, the most abundantly SUMOylated site, K77 showed a distinct transient peak in the S-phase and DSB time points (*Figure 7F*). K77 and the adjacent K79 are known sites of acetylation and their mutation disrupts silencing of telomeric and rDNA loci (*Hyland et al., 2005*). However, coincident SUMOylation and acetylation at K77 and K79 was not observed pointing to an antagonistic relationship at these sites. Possibly, K77/79 SUMOylation facilitates replication of transcriptionally repressed regions of the genome.

## Histone H1

H1 (Hho1) is a poorly conserved linker histone (*Millán-Ariño et al., 2016*) that binds DNA at nucleosome entry and exit points, and functions in chromatin compaction, silencing of rDNA, and suppression of homologous recombination (*Downs et al., 2003*; *Georgieva et al., 2012*; *Levy et al., 2008*; *Schäfer et al., 2008*). In budding yeast meiosis, H1 is inferred to help repress early meiotic genes, is subsequently depleted during prophase, and finally accumulates in mature spores to facilitate chromatin compaction (*Bryant et al., 2012*). We found H1 to be extensively SUMOylated, with levels increasing throughout meiosis. In particular, SUMO sites were scattered throughout the second DNA binding domain; and a prominent cluster was detected in the lysine-rich region that connects the DNA-binding domains (*Figure 7G*). Two of the highest intensity SUMO sites in this region, K131 and K133, were always present on a peptide co-modified with phosphorylation at S129, a CDK site (S-P). As with H2A, these SUMO sites do not conform to the canonical PDSM consensus but exhibit apparent dependence on phosphorylation. These observations point to a cooperative function for SUMO and phosphorylation in regulating H1 dynamics on meiotic chromatin.

In summary, histones are extensively SUMOylated during meiosis suggesting competitive and cooperative functions with other modifications to regulate gene expression, S-phase, DSB formation, recombination, and chromatin compaction.

## Histone modifiers and the SIR complex

Dynamic changes in meiotic histone modifications were accompanied by extensive SUMOylation of histone modifying enzymes, including acetyltransferases (SAGA, NuA3, Nua4 and ADA complexes), deacetylases (Set3C, Hda1, Rpd3S, Rpd3L, SIR, RENT, and Sum1-Rfm1-Hst1 complexes), the COMPASS methytransferase and the JmjC demethylase (*Supplementary file 4*). Here, we focus on the Sir (Silent Information Regulator) proteins, given their central roles in meiotic chromosome metabolism. Sir proteins are implicated in the pachytene-checkpoint response, crossover interference and global patterning of DSBs, including suppression of DSB formation in the rDNA array and sub-telomere regions (*Mieczkowski et al., 2007*; *Subramanian and Hochwagen, 2014*; *Vader et al., 2011*; *Zhang et al., 2014*).

The H4-K16Ac deacetylase Sir2 is recruited to rDNA as part of the RENT complex (regulator of nucleolar silencing and telophase exit), Cdc14-Net1-Sir2 (*Gartenberg and Smith, 2016*). Net1 is the key scaffolding protein for both assembly of RENT and its recruitment to the two rDNA non-transcribed spacers via interactions with Pol I and Fob1 (*Figure 7H*). These proteins represent a heavily SUMOylated cohort, with 16 sites on Net1, eight on Fob1, three on Sir2, a single site on Cdc14, and 20 sites across the nine subunits of RNA Pol I. Net1 SUMOylation intensity diminished during S-phase, increased again through the dHJ/synapsis time point, and then dropped again during CO formation (*Figure 7I*). This could reflect dynamic changes associated with rDNA replication and perhaps the release of Sir2 from the nucleolus to perform nuclear functions, such as the synapsis checkpoint and crossover interference. In vegetative cells, localization of Sir2 to the nucleolus is dependent on its SUMOylation (*Hannan et al., 2015*). However, SUMOylation of Sir2, Fob1 and two

prominently SUMOylated components of RNA Pol I (Rpa34 and Rpo26) all decreased as meiosis progressed (*Figure 7I*). Additionally, nucleolar localization of Sir2 during meiosis requires Dot1 (*San-Segundo and Roeder, 2000*), which is required for pachytene checkpoint arrest. These considerations suggest that the regulation of Sir2 dynamics may be distinct in meiotic versus mitotically cycling cells. rDNA stability also requires anchoring to the nuclear periphery, which involves another cohort of SUMOylated proteins that couple Fob1 to the membrane associated CLIP complex (*Gartenberg and Smith, 2016*). These include the Net1 paralog, Tof2 (with 15 acceptor sites) and the cohibin complex Csm1-Lrs4 (3 and 17 sites, respectively; *Figure 7J*). During mid meiosis, cohibin relocalizes to kinetochores and functions with meiosis-specific Mam1 as the monopolin complex, which mediates monopolar attachment of sister-kinetochores to the MI spindle (*Rabitsch et al., 2003*). Perhaps reflecting these dynamics, SUMOylation of Csm1 and Lrs4 was high through the G0, S and DSB time points, but then declined (*Figure 7K*). Mam1 was also SUMOylated on three sites, possibly functioning to stabilize monopolin (*Supplementary file 4*). In addition, Csm1-K139-SUMO could influence partner interactions as it lies close to the binding sites for the isopeptidase Ulp2 (which helps maintain silencing) (*Liang et al., 2017*) and Mam1.

To silence expression at locations other than the rDNA, Sir2 partners with Sir1, 3 and 4, which showed heterogeneous meiotic SUMO profiles suggesting distinct regulation (*Figure 7L*). Siz2-dependent SUMOylation of Sir2 disrupts its interaction with scaffold protein Sir4, causing Sir2 to redistribute from telomeres to the nucleolus (*Hannan et al., 2015*). In meiosis, Sir2 SUMOylation was relatively high in G0 and S, but dipped at the DSB time point and then remained relatively low suggesting that Sir2 largely redistributes to the nucleolus after S phase (*Figure 7L*). Sir4 was one of the most abundantly SUMOylated meiotic proteins with 30 acceptor sites in three apparent clusters that may differentially affect binding to DNA and histone-tails, and its interactions with partners such as Ku70/80, Rap1, Esc1, Sir2, and Sir3 (*Gartenberg and Smith, 2016*; *Figure 7M*). Sir4 SUMOylation increased throughout meiotic prophase roughly in parallel with partners that help recruit it to locations on chromatin (Sir1, Orc1 and Rap1), and to the nuclear envelope (Esc1; *Figure 7N*).

Sir3 binds nucleosomes to facilitate the assembly of silent chromatin domains (*Gartenberg and Smith, 2016*). In contrast to Sir4, SUMOylation of Sir3 showed a sharp peak during in S phase, was very low in the dHJ/synapsis time point and then increased again as crossovers formed (*Figure 7L and O*). Interestingly, this profile matches closely with that of Asf2, a poorly characterized Sir4 paralog whose overexpression disrupts silencing (*Buchberger et al., 2008*; *Le et al., 1997*). Our data suggest that Sir3 may partner Asf2 during meiosis. The general spike in SUMOylation of the SIR complex and its partners at the CO time point suggests late roles, perhaps in preparing chromosomes for segregation.

## Chromatin remodelers

Chromatin remodelers have well characterized roles in DSB repair in mitotically cycling cells (*Seeber et al., 2013*), but their roles in meiotic HR are less well characterized. Analysis in *Schizosaccharomyces pombe* invokes roles for multiple remodelers in meiotic HR, including Ino80, Swr1 and Fun30, and the Nap1 and Hir complex histone chaperones (*Storey et al., 2018*). Notably, in budding yeast, Chd1 was recently shown to act at the dHJ resolution step to facilitate MutLγ-dependent crossing over (*Wild et al., 2019*). All four families of chromatin remodelers in budding yeast (SWI/SNF, ISWI, CHD, INO80/SWRI), covering eight separate complexes (Ino80, Rsc, Swr1, Swi/Snf, Isw1a, Isw1b, Chrac, and Paf1), and the Hir complex were SUMOylated on multiple subunits during meiosis (*Supplementary file 4*). Consistent with a role for SUMOylation in activating the resolution function of Chd1, modification at an N-terminal cluster of four sites occurred only after $P_{GAL}$-*NDT80* expression, as dHJs were being resolved into crossovers (*Figure 7P*). Cox et al. showed that interaction between mammalian INO80 subunits, INO80E and TFPT, is mediated by SUMO (*Cox et al., 2017*). We detected SUMOylation on 11/15 yeast INO80 subunits. 10/14 SWR complex subunits were SUMOylated with a U-shaped SUMOylation profile (exemplified by Swc3 and Vps72; *Figure 7Q*). The drop in SUMOylation during S-phase is reminiscent of transcription factors such as Gcn4, suggesting that SUMOylation may suppress their activities. By contrast, bromodomain factor Bdf1, the SWR subunit with an affinity for acetylated H4 (*Matangkasombut and Buratowski, 2003*), showed a sharp transient SUMOylation peak during DSB formation (*Figure 7R*). Taf14 – a subunit shared

between TFIID, TFIIF, INO80, Swi/Snf, and NuA3m with an affinity for acetylated H3 – showed a similar SUMOylation peak suggesting roles in DSB formation or processing.

### Transcription factors

A majority of SUMOylated transcription factors (31 out of 37) showed a sharp decline in modification level during the transition from G0 to S, consistent with a repressive role for SUMO prior to meiotic entry. Sum1, a component of the Sum1-Rfm1-Hst1 complex that represses middle-meiotic genes prior to Ndt80 expression (*Winter, 2012*), showed a more progressive decline of SUMOylation suggesting a role in anticipation of Ndt80-dependent induction (*Figure 7S*). Coordinated SUMOylation of the Tup1-Ssn6 repressor and Gcn4 transcription factor hastens removal of Gcn4 to enable repression (*Ng et al., 2015*; *Rosonina et al., 2012*; *Texari et al., 2013*). Gcn4 is inferred to influence around a third of meiotic recombination hotspots and deletion of *GCN4* reduces recombination (*Abdullah and Borts, 2001*). SUMOylation of Tup1 and Gcn4 declined after G0 (*Figure 7T*). Notably, Gcn4-SUMO was lowest at the time of DSB formation and remained diminished thereafter suggesting that SUMO may influence the global DSB landscape via modification of transcription factors.

In summary, our mass spectrometry dataset marks a breakthrough for the field, delineating a diverse and dynamic meiotic SUMO-modified proteome and providing a rich resource for functional analyses, to identify pertinent targets and define regulatory networks. With this unique dataset in hand, major advances in understanding how SUMO regulates meiotic prophase are anticipated.

## Materials and methods

### Key resources table

| Reagent type (species) or resource | Designation | Source or reference | Identifiers | Additional information |
|---|---|---|---|---|
| Other | Histrap FF 1 ml | GE Healthcare | 17-5319-01 | AKTA FPLC cartridge |
| Peptide, recombinant protein | Lysyl Endopeptidase (Lys-C) | Wako Chemicals | 125–02541 | |
| Peptide, recombinant protein | Glu-C, sequencing grade | Promega | V1651 | |
| Other | Sep-Pak tC18 1cc vac cartridge 50 mg | Waters | WAT054960 | Desalting cartridge |
| Commercial assay, kit | PTMScan Ubiquitin Remnant Motif (K-ε-GG) | Cell Signaling technology | 5562S | Antibody beads and buffer kit |
| Chemical compound, drug | PP1 | Tocris | 1397 | Synthetic ATP analog |
| Chemical compound, drug | 3-Indoleacetic acid (Auxin) | Sigma-Aldrich | I3750-25G-A | Protein alkylation agent |
| Chemical compound, drug | β-Estradiol | Sigma-Aldrich | E2758 | |
| Antibody | Anti-c-Myc 'mouse monoclonal' antibody | Roche | 11667149001, RRID:AB_390912 | '1:5000' |
| Antibody | Anti-Arp7 'goat polyclonal' antibody | Santa Cruz Biotechnology | y-C20, RRID:AB_671730 | '1:2000' |
| Antibody | 'Donkey polyclonal' anti-goat, IRDye 680 | LI-COR Biosciences | 926–68074, RRID:AB_10956736 | '1:5000' |
| Antibody | 'Donkey polyclonal' anti-mouse, IRDye 800 | LI-COR Biosciences | 926–32212, RRID:AB_621847 | '1:5000' |

*Continued on next page*

*Continued*

| Reagent type (species) or resource | Designation | Source or reference | Identifiers | Additional information |
|---|---|---|---|---|
| Antibody | 'Goat polyclonal' anti-guinea pig, Alexa Fluor 555 | Thermo Fisher | A-21435, RRID:AB_2535856 | '1:200' |
| Antibody | Anti-Zip1 'goat polyclonal' | Santa Cruz Biotechnology | y-N16, RRID:AB_794259 | '1:50' |
| Antibody | Anti-Zip1 'guinea pig polyclonal' | Dr. Scott Keeney | gp3 | '1:500' |
| Antibody | 'Donkey polyclonal' anti-goat, Alexa Fluor 555 | Thermo Fisher | A-21432, RRID:AB_141788 | '1:1000' |
| Software, algorithm | MaxQuant 1.6.1 | Max Planck Institute | RRID:SCR_014485 | MS raw data search |

## Strain table

| Strain number | Mating type | Genotype | Experiment |
|---|---|---|---|
| NHY7590 | a/α | NHY7067 x NHY7083 | SUMO Proteomics |
| NHY10438 | a/α | NHY10416 x NHY10425 | Aos1-AID Condition 1 |
| NHY10439 | a/α | NHY10419 x NHY10422 | Aos1-AID Condition 1 |
| NHY10406 | a/α | NHY10318 x NHY10394 | Aos1-AID Condition 2,3 |
| NHY10407 | a/α | NHY10319 x NHY10395 | Aos1-AID Condition 2,3 |
| NHY10408 | a/α | NHY10316 x NHY10396 | Aos1-AID Condition 2,3 |
| NHY10413 | a/α | NHY10402 x NHY10410 | Aos1-AID Condition 4 |
| NHY10414 | a/α | NHY10403 x NHY10411 | Aos1-AID Condition 4 |
| NHY10415 | a/α | NHY10398 x NHY10412 | Aos1-AID Condition 4 |
| NHY10440 | a/α | NHY10428 x NHY10436 | Uba2-AID Condition 1 |
| NHY10441 | a/α | NHY10431 x NHY10434 | Uba2-AID Condition 1 |
| NHY10371 | a/α | NHY10320 x NHY10366 | Uba2-AID Condition 2,3 |
| NHY10372 | a/α | NHY10321 x NHY10368 | Uba2-AID Condition 2,3 |
| NHY10382 | a/α | *NHY10380 x NHY10381* | Uba2-AID Condition 4 |
| NHY10383 | a/α | *NHY10378 x NHY10381* | Uba2-AID Condition 4 |
| NHY7067 | a | *ho LYS2 HIS4::LEU2-(BamHI; +ori) HIS6-StrepII-Smt3-KGG::KanMX ura3::pGPD1-GAL4-ER::URA3 pGAL-Ndt80::TRP1 irt1::pCUP1-IME1::NatMX ime4::pCUP1-IME4::NatMX* | |
| NHY7083 | α | *ho his4-X::LEU2-(NgoMIV; +ori)–URA3 HIS6-StrepII-Smt3-KGG::KanMX ura3::pGPD1-GAL4-ER::URA3 pGAL-Ndt80::TRP1 irt1::pCUP1-IME1::NatMX ime4::pCUP1-IME4::NatMX* | |
| NHY10416 | α | *ho::hisG leu2::hisG ura3(ΔSma-Pst) HIS4::LEU2-(BamHI; +ori) pCUP1-1-OsTIR1-9Myc-URA3 AOS1-AID-9MYC::HphMX* | |
| NHY10419 | a | *ho::hisG leu2::hisG ura3(ΔSma-Pst) HIS4::LEU2-(BamHI; +ori) pCUP1-1-OsTIR1-9Myc-URA3 AOS1-AID-9MYC::HphMX* | |
| NHY10422 | α | *ho::hisG leu2::hisG ura3(ΔSma-Pst) his4-X::LEU2-(NgoMIV; +ori)–URA3 lys2::HphMX::pCUP1-1-OsTIR1-9myc AOS1-AID-9MYC::HphMX* | |
| NHY10425 | a | *ho::hisG leu2::hisG ura3(ΔSma-Pst) his4-X::LEU2-(NgoMIV; +ori)–URA3 lys2::HphMX::pCUP1-1-OsTIR1-9myc AOS1-AID-9MYC::HphMX* | |
| NHY10316 | α | *ho leu2::hisG arg4-Nsp lys2 ura3(ΔSma-Pst) HIS4::LEU2-(BamHI; +ori) pCUP1-1-OsTIR1-9Myc-URA3 cdc7-as3-MYC mcm5-bob1 AOS1-AID-9MYC::HphMX* | |

*Continued*

| Strain number | Mating type | Genotype | Experiment |
|---|---|---|---|
| NHY10318 | a | ho leu2::hisG arg4-Nsp lys2 ura3(ΔSma-Pst) HIS4::LEU2-(BamHI; +ori) pCUP1-1-OsTIR1-9Myc-URA3 cdc7-as3-MYC mcm5-bob1 AOS1-AID-9MYC::HphMX | |
| NHY10319 | a | ho leu2::hisG arg4-Nsp lys2 ura3(ΔSma-Pst) HIS4::LEU2-(BamHI; +ori) pCUP1-1-OsTIR1-9Myc-URA3 cdc7-as3-MYC mcm5-bob1 AOS1-AID-9MYC::HphMX | |
| NHY10394 | α | ho::hisG leu2::hisG ura3(ΔSma-Pst) his4-X::LEU2-(NgoMIV; +ori)–URA3 lys2::HphMX::pCUP1-1-OsTIR1-9myc cdc7-as3-MYC mcm5-bob1 AOS1-AID-9MYC::HphMX | |
| NHY10395 | α | ho::hisG leu2::hisG ura3(ΔSma-Pst) his4-X::LEU2-(NgoMIV; +ori)–URA3 lys2::HphMX::pCUP1-1-OsTIR1-9myc cdc7-as3-MYC mcm5-bob1 AOS1-AID-9MYC::HphMX | |
| NHY10396 | a | ho::hisG leu2::hisG ura3(ΔSma-Pst) his4-X::LEU2-(NgoMIV; +ori)–URA3 lys2::HphMX::pCUP1-1-OsTIR1-9myc cdc7-as3-MYC mcm5-bob1 AOS1-AID-9MYC::HphMX | |
| NHY10398 | α | ho::hisG leu2::hisG ura3(ΔSma-Pst) HIS4::LEU2-(BamHI; +ori) HphMX::PGAL1-NDT80 pCUP1-1-OsTIR1-9Myc-URA3 AOS1-AID-9MYC::HphMX | |
| NHY10402 | a | ho::hisG leu2::hisG ura3(ΔSma-Pst) HIS4::LEU2-(BamHI; +ori) HphMX::PGAL1-NDT80 pCUP1-1-OsTIR1-9Myc-URA3 AOS1-AID-9MYC::HphMX | |
| NHY10403 | a | ho::hisG leu2::hisG ura3(ΔSma-Pst) HIS4::LEU2-(BamHI; +ori) HphMX::PGAL1-NDT80 pCUP1-1-OsTIR1-9Myc-URA3 AOS1-AID-9MYC::HphMX | |
| NHY10410 | α | ho::hisG leu2::hisG ura3(ΔSma-Pst) his4-X::LEU2-(NgoMIV; +ori)–URA3 Hygro::PGAL1-NDT80 ura3:PGPD1-GAL4(848)-ER:URA3 lys2::HphMX::pCUP1-1-OsTIR1-9myc AOS1-AID-9MYC::HphMX | |
| NHY10411 | α | ho::hisG leu2::hisG ura3(ΔSma-Pst) his4-X::LEU2-(NgoMIV; +ori)–URA3 HphMX::PGAL1-NDT80 ura3:PGPD1-GAL4(848)-ER:URA3 lys2::HphMX::pCUP1-1-OsTIR1-9myc AOS1-AID-9MYC::HphMX | |
| NHY10412 | a | ho::hisG leu2::hisG ura3(ΔSma-Pst) his4-X::LEU2-(NgoMIV; +ori)–URA3 Hygro::PGAL1-NDT80 ura3:PGPD1-GAL4(848)-ER:URA3 lys2::HphMX::pCUP1-1-OsTIR1-9myc AOS1-AID-9MYC::HphMX | |
| NHY10428 | α | ho::hisG leu2::hisG ura3(ΔSma-Pst) HIS4::LEU2-(BamHI; +ori) pCUP1-1-OsTIR1-9Myc-URA3 UBA2-AID-9MYC::HphMX | |
| NHY10431 | a | ho::hisG leu2::hisG ura3(ΔSma-Pst) HIS4::LEU2-(BamHI; +ori) pCUP1-1-OsTIR1-9Myc-URA3 UBA2-AID-9MYC::HphMX | |
| NHY10434 | α | ho::hisG leu2::hisG ura3(ΔSma-Pst) his4-X::LEU2-(NgoMIV; +ori)–URA3 lys2::HphMX::pCUP1-1-OsTIR1-9myc UBA1-AID-9MYC::HphMX | |
| NHY10436 | a | ho::hisG leu2::hisG ura3(ΔSma-Pst) his4-X::LEU2-(NgoMIV; +ori)–URA3 lys2::HphMX::pCUP1-1-OsTIR1-9myc UBA1-AID-9MYC::HphMX | |
| NHY10320 | a | ho leu2::hisG arg4-Nsp lys2 ura3(ΔSma-Pst) HIS4::LEU2-(BamHI; +ori) pCUP1-1-OsTIR1-9Myc-URA3 cdc7-as3-MYC mcm5-bob1 UBA2-AID-9MYC::HphMX | |
| NHY10321 | a | ho leu2::hisG arg4-Nsp lys2 ura3(ΔSma-Pst) HIS4::LEU2-(BamHI; +ori) pCUP1-1-OsTIR1-9Myc-URA3 cdc7-as3-MYC mcm5-bob1 UBA2-AID-9MYC::HphMX | |
| NHY10366 | α | ho::hisG leu2::hisG ura3(ΔSma-Pst) his4-X::LEU2-(NgoMIV; +ori)–URA3 lys2::HphMX::pCUP1-1-OsTIR1-9myc cdc7-as3-MYC mcm5-bob1 UBA2-AID-9MYC::HphMX | |
| NHY10368 | α | ho::hisG leu2::hisG ura3(ΔSma-Pst) his4-X::LEU2-(NgoMIV; +ori)–URA3 lys2::HphMX::pCUP1-1-OsTIR1-9myc cdc7-as3-MYC mcm5-bob1 UBA2-AID-9MYC::HphMX | |
| NHY10377 | a | ho::hisG leu2::hisG ura3(ΔSma-Pst) HIS4::LEU2-(BamHI; +ori) HphMX::PGAL1-NDT80 pCUP1-1-OsTIR1-9Myc-URA3 UBA2-AID-9MYC::HphMX | |
| NHY10380 | α | ho::hisG leu2::hisG ura3(ΔSma-Pst) HIS4::LEU2-(BamHI; +ori) HphMX::PGAL1-NDT80 pCUP1-1-OsTIR1-9Myc-URA3 UBA2-AID-9MYC::HphMX | |
| NHY10381 | a | ho::hisG leu2::hisG ura3(ΔSma-Pst) his4-X::LEU2-(NgoMIV; +ori)–URA3 HphMX::PGAL1-NDT80 ura3:PGPD1-GAL4(848)-ER:URA3 lys2::HphMX::pCUP1-1-OsTIR1-9myc UBA2-AID-9MYC::HphMX | |

## Experimental model

### Yeast strains and plasmids for SUMO proteomics

The yeast SUMO gene, *SMT3* was N-terminally tagged as described previously (*Booher and Kaiser, 2008*), with some modifications. Plasmid pFA6a-kanMX6-$P_{GAL1}$-HBH (a gift from Dr. Peter Kaiser) was modified by replacing the HBH tag module with a 6His-Strep-tagII module. A PCR product containing *kanMX6-$P_{GAL1}$-6His-StrepII*, with 70 bp homologies to the 5' region of *SMT3* was used to

transform haploid *GAL+* yeast. Transformants were selected on YP-Gal + Geneticin (G418) solid media and genotyped by PCR. To restore the native *SMT3* promoter, these strains were transformed with $P_{SMT3}$ and selected for growth on YPD. Strains were then crossed to introduce $Cu^{2+}$-inducible versions of *IME1* and *IME4* (a gift from Folkert van Werven and Angelika Amon) and an estradiol-inducible *NDT80* gene (*Benjamin et al., 2003*), which allowed for inducible, highly synchronous meiotic entry and pachytene exit, respectively. To generate the I96K mutation in Smt3, strains were transformed with a PCR product generated from *pFA6a-3HA-kanMX6* (*Longtine et al., 1998*) with PCR primers Smt3 KGG FW (*AAGATTTGGACATGGAGGATAACGATATTATTGAGGCTCACAGAGAACAGAAAGGTGGTGCTACGTATTAGCGGATCCCCGGGTTAATTAA*) and R1 Smt3 (*CTATGGTATTTTTTGGTGGGTGGAGGGAAGGGAGAGGTTTGTTGGCGTTCTTTAGGCATTGTTAAGAGTCGAATTCGAGCTCGTTTAAAC*). Transformants were screened by PCR genotyping and validated by sequencing. Integrity of the *HIS4::LEU2* hotspot was validated by Southern blotting. Strains of opposite mating types were mated to generate the diploid strain used in the proteomic experiments (Strain Table).

## Yeast strains for DNA physical assays

Full genotypes are shown in the strain table. The Auxin-Induced Degron (AID) system designed for use during meiosis has been described (*Tang et al., 2015*). Minimal AID fusions to Aos1 and Uba2 were constructed as described using plasmid p7aid-9m as a template for PCR-mediated allele replacement (*Morawska and Ulrich, 2013*; *Tang et al., 2015*). The estradiol-inducible *GAL4-ER IN-NDT80* system has been described (*Benjamin et al., 2003*; *Carlile and Amon, 2008*; *Louvion et al., 1993*; *Tang et al., 2015*).

## Method details

### Meiotic time courses for proteomic analysis

Synchronous meiotic time courses were performed as described (*Owens et al., 2018*) with the following modifications. Saturated 5 mL overnight YPD cultures were diluted 1:1000 into six 300 ml SPS in 2 L flasks and shaken overnight at 325 rpm. At 18 hr, cells were collected by centrifugation, washed with sterilize water and divided into twelve 2.8L baffled flasks, each containing 350 mL SPM. These cultures were then shaken at 350 rpm for two hours before adding $CuSO_4$ to a final concentration of 25 µM. Cells were collected at 0 (G0), 1.5 (premeiotic S-phase, S), 2.5 (double-strand breaks, DSB), 3.5 (strand invasion and synapsis, SI), and 5 hr after $CuSO_4$ addition (double Holliday junction formation and pachytene, dHJ). At the 5 hr time point, estradiol was added to a final concentration of 1 µM and a final sample was collected at 6 hr (crossing over and SC disassembly, CO). For each timepoint, cells from all 12 flasks were pooled, pelleted by centrifugation, washed with milliQ water, pelleted again and then flash frozen in liquid nitrogen. Frozen pellets were stored at −80℃ until all the samples had been collected, and then processed in parallel. For all quantitative analyses that look at the temporal changes in SUMOylations, a set of triplicate time courses were collected in order to account for inter-sample variation, as well as missing values in proteomics data. These were then processed together to reduce processing variations.

## Cell lysis and protein purification for proteomics

Each 20–25 g cell pellet was resuspended in 60 mL fresh ice-cold 0.25 M NaOH, 1% β-mercaptoethanol lysis solution and incubated on ice for 20 min. Proteins were precipitated with 10 mL 100% trichloroacetic acid on ice for 20 min and pelleted by centrifugation at 5000 x g for 15 min. The pellet was broken up, thoroughly washed twice with 5 ml ice-cold acetone and air dried. Proteins were dissolved in 6 M Guanidine buffer (6 M Guanidine, 100 mM Tris pH 8.0, 500 mM NaCl, 10 mM imidazole pH 8.0) by vigorous shaking at 30℃. Lysates were clarified by centrifugation at 10,000 x g for 30 min and filtration through a Whatman filter paper. A 1 mL HisTrap FF column mounted on an Äkta Avant FPLC system (GE Lifesciences) was equilibrated with 5 mL of 6 M Guanidine buffer and then the cell lysate was passed through, followed by washes with 20 mL 6 M Guanidine buffer, 10 mL each of Urea buffer pH 6.3 (6 M urea, 100 mM tris pH 6.3, 500 mM NaCl), and Urea buffer pH 8.0 (6 M urea, 100 mM Ammonium Bicarbonate pH 8.0, 150 mM NaCl, 20 mM Imidazole). Bound proteins were eluted with 0.5 M Imidazole (6 M urea, 100 mM Ammonium Bicarbonate pH 8.0, 150 mM NaCl, 500 mM Imidazole pH 8.0) and collected in 2 mL fractions. Fractions covering the elution peak

were pooled and final protein concentrations were determined by the Bradford assay. Proteins were reduced with 4 mM TCEP (Pierce) for 30 min at room temperature, alkylated with 10 mM iodoaceta-mide (Sigma) for 30 min in the dark and then quenched with 10 mM DTT. There is some concern about iodoacetamide producing artifacts that can be mistaken for a diGly modification (Nielsen et al., 2008). However, in a control experiment where a second anti-diGly immuno-enrich-ment step was not employed, we failed to detect any diGly signatures (data not shown), showing that this concern is not warranted in this case. A total of 2 mg protein from each sample was trans-ferred to a new tube, samples were diluted with elution buffer so that all samples had the same final volume and then diluted with twice that volume of 100 mM ammonium bicarbonate pH 8.0. Samples were digested with 0.4 AU Lys-C (Wako Chemicals) overnight at 37°C on a shaker at 225–250 rpm. The next day, samples were split and one half was diluted with 100 mM ammonium bicarbonate to 0.8 M urea and digested for another 6 hr with 10 μg Glu-C. Digestion was stopped by acidification with 10% trifluoroacetic acid (TFA) to pH ≤3, and samples were the desalted on Sep-Pak tC18 reversed phase columns (Waters). The desalted samples were lyophilized for at least 48 hr and then dissolved in 500 μL 1X immunoaffinity purification buffer (IAP buffer, Cell Signaling Technologies). Samples were sonicated for 30 min at room temperature in a water-bath sonicator and then clarified by centrifugation. Peptides containing di-glycyl lysine (K-ε-GG) residues were enriched using the UbiScan kit (Cell Signaling technologies) according to the manufacturer's instructions with some modifications. Each tube of immunoaffinity beads was equilibrated with the IAP buffer and then split evenly into six tubes. Each set of six tubes was used for one set of timepoints (i.e. one tube per time course) to reduce variation. Peptide solutions were incubated with UbiScan beads at 4°C for 90 min, washed twice with chilled IAP buffer and twice with chilled HPLC grade water (Fisher). Bound pepti-des were then eluted twice with 55 μL 0.15% TFA for 5–10 min at room temperature, eluates were pooled, flash frozen and then dried by vacuum centrifugation.

## Mass spectrometry

K-ε-GG enriched peptide mixtures were analyzed with a Q Exactive Orbitrap tandem MS system (Thermo Scientific) with an upstream in-line Proxeon Easy-nLCII HPLC system (Thermo Scientific). Peptides were resuspended in 0.2% TFA and loaded on to a 25 mm Magic C18 RPLC column and eluted over a 90 min acetonitrile gradient at 300 nl/min. MS1 spectra were sampled with a top-20 cutoff and 5 s dynamic exclusion and subjected to high-energy collision dissociation to obtain MS2 spectra. For the triplicate set that was used for quantitative analysis, all samples were run back-to-back on the same LC column, with the same instrument settings to minimize variation. Additionally, the run-order of samples was randomized to eliminate any effect on the final data.

## Time course conditions for SUMO execution-point analysis

### Auxin-induced degradation of SUMO E1 proteins

As described (Tang et al., 2015), 50 μM CuSO$_4$ (diluted from 100 mM stock) was added to induce expression of $P_{CUP1}$-OsTIR1 and 2 mM auxin (dilution from a 2 M stock of 3-indoleacetic acid, Sigma I3750, dissolved in DMSO) was added to trigger Aos1-AID or Uba2-AID degradation.

### cdc7-as3 time courses

Cells were incubated in SPM as previously described (Oh et al., 2009). After 30 min, PP1 (Tocris Bio-science 1397, 10 mM stock dissolved in DMSO) was added for a final concentration of 3.5 μM. This concentration is sufficient to block DSB formation. After 6.5 hr in PP1, cells were released from arrest by washing nine times in an equal volume of sporulation media over a 90 min period.

### To assay DSB formation following E1 degradation

Cells were cultured as for cdc7-as3 time courses. At 5.5 hr, cultures were split and at 6 hr copper (1:2000 dilution of 100 mM CuSO4 stock for a final concentration of 50 μM) and auxin (1:1000 dilu-tion of 2 M stock for a final concentration of 2 mM) were added to one sub-culture. At 6.5 hr, cul-tures were washed as described above with media containing copper and auxin. At 8 hr, cultures were placed in fresh flasks and at 8.5 hr, auxin was added again (1:2000 dilution of 2 M stock). Cell samples were then collected to assay protein depletion, meiotic divisions, and recombination intermediates.

### To assay recombination intermediates following E1 degradation

Cells were cultured as for *cdc7-as3* time courses. After washes were completed at 8 hr, cultures were split and at 9 hr, copper and auxin were added to one subculture. At 9.5 hr, auxin was added again (1:2000 dilution of 2 M stock). Cell samples were then collected to assay protein depletion, meiotic divisions, and recombination intermediates.

### IN-NDT80 time courses

Cells were incubated in SPM as previously described (*Oh et al., 2009*). At 6.5 hr, copper was added, and at 7 hr auxin was added to one subculture, and 1 µM estradiol (Sigma E2758 in DMSO) was added to both subcultures to induce *IN-NDT80.* At 7.5 hr, auxin was added again (1:2000 dilution of 2 M stock). Cell samples were collected to assay protein depletion, meiotic divisions, and recombination intermediates.

### Southern blot analysis of the DNA events of meiosis

Detailed protocols for meiotic time courses and DNA physical assays at the *HIS4::LEU2* recombination hotspot have been described (*Oh et al., 2009*). Error bars show averages (± SD) from three experiments.

### Immunblotting

Whole cell extracts were prepared using a TCA extraction method, as described (*Johnson and Blobel, 1999*; *Tang et al., 2015*). Following SDS-PAGE and western blotting, an anti-c-Myc mouse monoclonal antibody (Roche; 11667149001) was used to detect Aos1-AID-9myc and Uba2-AID-9Myc. Anti-Arp7 goat polyclonal antibody (Santa Cruz Biotechnology; y-C20) was used to detect Arp7 as a loading control. Donkey anti-goat (IRDye 680; LI-COR Biosciences; 926–68074) and donkey anti-mouse (IRDye 800; LI-COR Biosciences; 926–32212) were used as secondary antibodies. Membranes were imaged using a LI-COR Odyssey system.

### Surface spreading of meiotic nuclei, immunofluorescence, and quantification of synapsis

Surface spreading of meiotic nuclei was performed as described by *Grubb et al., 2015*. During spheroplasting, 20 µl of dithiothreitol (DTT) was used instead of the published 40 µl. Fixation was achieved with 4% PFA/sucrose solution. Spreads from *Figure 3J,M* were stained with anti-Zip1 guinea pig polyclonal antibody (a gift from Scott Keeney, 1:500 dilution), and then with goat anti-guinea pig polyclonal secondary antibody (Alexa Fluor 555; ThermoFisher; A-21435, 1:200 dilution). Spreads from *Figure 3—figure supplement 2J* were stained with anti Zip1 goat polyclonal antibody (Santa Cruz Biotechnology; y-N16, 1:50 dilution), and then with donkey anti-goat polyclonal secondary antibody (Alexa Fluor 555; ThermoFisher; A-21432, 1:1000 dilution). Slides were mounted in antifade with added DAPI (Prolong Gold, Invitrogen; P36930) and images were captured using a Zeiss AxioPlan II microscope, Hamamatsu ORCA-ER CCD camera and Volocity software. Synapsis was quantified by characterizing four classes of Zip1 staining pattern: no Zip1, foci only, a mixture of foci and lines, and lines only (as previously described by *Chen et al., 2015*).

### FACS analysis

Cell pellets from 200 µl of meiotic culture were fixed in 70% ethanol. Cells were then washed in 1 ml of 50 mM sodium citrate (pH 7.5) and resuspended in 1 ml of 50 mM sodium citrate (pH 7.5) with 130 µg RNaseA and incubated at 37°C for 1 hr. 0.52 mg of Proteinase K was added and samples were incubated at 65°C for an additional hour. 100 µl of sodium citrate (pH 7.5) containing a 1:10,000 dilution of SYBR Green (Invitrogen S7563) and 25 µl of 10% Triton-X 100 were added, and cells were sonicated with a probe sonicator on setting 1.5 for 10 s. Cells were scanned on a FACScan (BD Biosciences) and data was acquired with CellQuest Pro, using the FL1 (green) detector to trigger doublet discrimination. Data was analyzed using FlowJo to gate live, single-cell events measuring SYBR Green signal. Histograms were plotted modally, or as the percent of the maximum.

## Quantification and statistical analysis

### Processing of mass spectrometry data

Raw files were searched with MaxQuant 1.6.1 (Max Planck Institute) (*Cox and Mann, 2008*). Samples from complementary digests for the same replicate were processed under the same experiment heading. Two digestion (Lys-C only and Lys-C + Glu-C) strategies were processed as separate groups. For all samples, a maximum of five missed cleavages and potential modifications were allowed per peptide. Two separate searches were run. Protein N-terminal acetylation, methionine oxidation, and di-glycyl lysine were set as variable modifications, and carbamidomethyl cysteine was set as fixed modification in both runs. In addition, one search included serine, threonine and tyrosine phosphorylations, and the other had lysine acetylation as variable modifications. For peptides, a minimum length of 6 and maximum mass of 5200 Da were allowed. Label-free quantification was enabled with default settings (*Cox et al., 2014*), and was applied to all variable modifications. Standard orbitrap settings were used under the 'instrument' parameter. An *S. cerevisiae* proteome downloaded from Uniprot was used for searches. Search parameters were adjusted to search for second peptides and dependent peptides, and to match identifications between samples. Defaults were used for all settings unless specified otherwise. Data tables generated by MaxQuant were used for further analysis.

### Data analysis

All proteomics data were processed with Perseus (Max Planck Institute) (*Tyanova et al., 2016*). Text files titled 'proteinGroups' and 'GlyGly (K)Sites' were loaded into a Perseus workspace. Protein and modification matrices were filtered to remove reverse matches and potential modifications. SUMOylated proteins were identified by filtering protein matrices for 'GlyGly' modification. The list of Uniprot IDs of SUMOylated proteins was loaded on Panther Gene Ontology search (http://www.pantherdb.org/) and a statistical over-representation test was run for GO-slim Biological Processes and GO-slim cellular Component, with *S. cerevisiae* proteome as the background. To identify SUMOylation consensus sequence motifs, the motif configuration file of Perseus was edited to add the known SUMOylation consensus sequences, and motifs were added to the GlyGly-site matrix with 'sequence window' as the search target. To calculate the number of proteins and SUMO sites identified at each timepoint, protein and SUMO-site matrices from the triplicate dataset were exported to Microsoft Excel and the entries in identification type columns were converted to numbers. Numbers for each triplicate sample were added up and rows containing valid values were counted to give number of identified proteins and sites respectively.

### Quantitative proteomics

Quantitative analysis was carried out in Perseus, with label-free quantitation generated by MaxQuant. For proteins, the 'LFQ' values were used. For SUMO sites, 'intensity $x\_y$' values were used, where $x$ is the timepoint and $y$ is the replicate number. To analyze the temporal changes in SUMOylation states of proteins and SUMO sites, a new categorical annotation called 'time' was applied to the protein and site matrices, and individual replicates were labeled to identify the timepoint they belonged to. This categorical annotation was used to average the quantitative data for triplicate samples at each timepoint. These averages were used for further analysis. To visualize the changes in protein SUMOylation profiles across the time course, the averaged LFQ profiles were loaded into Morpheus (https://software.broadinstitute.org/morpheus) and a similarity matrix was generated. To generate hierarchical clustering, proteins were filtered to isolate those that were quantifiable for at least one timepoint. LFQ values were transformed to Log (2). Invalid values were imputed from a separate normal distribution for each column. All values were then normalized by Z-scoring to fit them in a range of −2 to +2. These normalized LFQs were loaded into Morpheus and the rows were subjected hierarchical clustering by 1- Pearson correlation to generate clustering tree and heatmap.

The cumulative intensities of all identified sites were retrieved from the 'intensity' values from the GlyGly site matrix of a full search of all (63) samples.

## Identifying types of secondary structure that are enriched for SUMOylation sites

For all proteins with at least one identified SUMOylation site, each lysine residue was scored for whether or not it was predicted to reside in different structural contexts. Globular domains, long disordered regions, and short disordered regions were predicted using IUPred (*Dosztányi, 2018*) with default settings. An additional 'Disordered' class was defined as any region predicted to be disordered, that is not inside a predicted globular domain. Solvent exposed and buried regions were predicted using ACCpro from the Scratch package (*Magnan and Baldi, 2014*). 'Globular-Exposed' and 'Globular-Buried' classes were then defined based on the combined IUPred and ACCpro predictions. Coiled-coil regions were predicted using ncoils with default parameters (*Lupas et al., 1991*). Transmembrane regions were predicted using HMMTOP version 2.1 with default settings (*Tusnády and Simon, 1998*). 'Disordered Not CC' was defined from the Disordered and coiled coil predictions. 'Disordered Terminus' was defined as a disordered region that extends to the N- or C-terminus of the protein. 'Disorder in 100 AA of Terminus' was defined as being in a Disordered Not CC region and being within 100 amino acids of either the N- or C-terminus of the protein. For each structure class, the fold-enrichment was computed as the percentage of detected lysine SUMOylation sites falling within the structure class divided by the percentage of all lysines (in proteins with at least one detected SUMOylation site) falling within the same structure class.

## Analysis of SUMOylation site clustering

For all proteins with at least one identified SUMOylation site, the fraction of lysines detected as SUMOylated was computed as a function of the distance in amino acids from either a SUMOylated lysine (distance from SUMO-K in *Figure 4F*) or from a lysine (distance from K). To compute these curves, for every SUMOylation site in our data set we computed the distance to each other lysine in the protein and annotated whether or not the lysine was detected as a SUMOylation site. We then pooled the data for all SUMOylation sites and computed the fraction of lysines that were detected as SUMOylation sites as a function of distance in amino acids (rounded to the nearest 10 amino acids). We then completed an analogous computation using all lysines rather than just the SUMOylation sites.

## Local disorder analysis around SUMOylation sites

We measured the distribution of quantitative 'Short Disorder' scores from IUPred for lysines detected as SUMOylation sites and for lysines that were not detected as SUMOylation sites.

## Diagrams of SUMOylation sites relative to protein domains and secondary structure

For all proteins with at least one identified SUMOylation site, we generated an image annotating the positions of all lysines (upper track, black lines), all detected SUMOylation sites (upper track, red lines with residue numbers annotated), all predicted SUMO-interacting motifs (SIMs) defined using the GPS-SUMO algorithm with either the high (upper track, dark gray rectangles), medium (upper track, gray rectangles), or low (upper track, light gray rectangles) score threshold (*Zhao et al., 2014*). In the middle track, protein domains detected with HMMER version 3.1b2 and Pfam-A version 29 set of models are shown (*Eddy, 1998*; *Finn et al., 2016*). In the bottom track, globular domains predicted by IUPred (blue rectangles), coiled-coil regions predicted by ncoils (red rectangles), and transmembrane regions predicted by HMMTOP (black rectangles) are shown. The short disorder score from IUPred is also indicated by the width of light red region.

## Acknowledgements

We thank Chris Lima (MSKCC), Kevin Corbett (UC San Diego), Katrin Karbstein (The Scripps Research Institute), Ilan Attali and James Berger (Johns Hopkins University School of Medicine), Mike Botchan (UC Berkeley), Wolf Heyer (UC Davis), Michael Lichten (CCR, NCI) and members of the Hunter Lab for support and discussions. Anthony Herren and Brett Phinney (UC Davis Proteomics Core) provided dedicated proteomics services. pFA6a-kanMX6-PGAL1-HBH was a gift from Peter Kaiser and Hongwei Zhang (UC Irvine). This work was supported by NIH NIGMS grant GM074223 to

NH SO was supported by NIH NIGMS T32 Training Program in Molecular and Cellular Biology 5T32GM007377 and an F31 Ruth L Kirschstein National Research Service Award 1F31GM125106. MI was supported by a JSPS postdoctoral fellowship. NH is an Investigator of the Howard Hughes Medical Institute.

## Additional information

### Funding

| Funder | Grant reference number | Author |
|---|---|---|
| National Institute of General Medical Sciences | GM074223 | Neil Hunter |
| Howard Hughes Medical Institute | Investigator Award | Neil Hunter |
| National Institute of General Medical Sciences | 5T32GM007377 | Shannon N Owens |
| National Institute of General Medical Sciences | 1F31GM125106 | Shannon N Owens |
| Japan Society for the Promotion of Science | postdoctoral fellowship | Masaru Ito |

The funders had no role in study design, data collection and interpretation, or the decision to submit the work for publication.

### Author contributions

Nikhil R Bhagwat, Conceptualization, Resources, Data curation, Formal analysis, Supervision, Validation, Investigation, Methodology, Writing - original draft, Project administration, Writing - review and editing; Shannon N Owens, Masaru Ito, Resources, Data curation, Formal analysis, Validation, Investigation, Methodology, Writing - original draft; Jay V Boinapalli, Resources, Data curation, Investigation, Writing - review and editing; Philip Poa, Alexander Ditzel, Srujan Kopparapu, Meghan Mahalawat, Data curation, Validation, Investigation, Writing - review and editing; Owen Richard Davies, Formal analysis, Visualization, Writing - review and editing; Sean R Collins, Software, Formal analysis, Visualization, Methodology, Writing - review and editing; Jeffrey R Johnson, Resources, Methodology, Writing - review and editing; Nevan J Krogan, Supervision, Methodology, Writing - review and editing; Neil Hunter, Conceptualization, Resources, Data curation, Formal analysis, Supervision, Funding acquisition, Investigation, Visualization, Methodology, Writing - original draft, Project administration, Writing - review and editing

### Author ORCIDs

Nikhil R Bhagwat ⓘD https://orcid.org/0000-0002-2945-6453
Shannon N Owens ⓘD http://orcid.org/0000-0002-0810-0116
Owen Richard Davies ⓘD http://orcid.org/0000-0002-3806-5403
Sean R Collins ⓘD http://orcid.org/0000-0002-4276-5840
Neil Hunter ⓘD https://orcid.org/0000-0003-1498-2327

### Decision letter and Author response

Decision letter https://doi.org/10.7554/eLife.57720.sa1
Author response https://doi.org/10.7554/eLife.57720.sa2

## Additional files

### Supplementary files

• Supplementary file 1. Comparison of SUMOylated proteins identified in this study with data from previous studies.

• Supplementary file 2. Key MaxQuant Tables. Compilation of useful tables generated by MaxQuant is a single combined search of all samples: ProteinGroups: All proteins including unmodified proteins, contaminants and false positives GlyGly sites: SUMO sites Phospho (STY) sites: phosphorylated serine, threonine and tyrosine residues ModificationSpecific Peptides: All identified peptides and their relevant modifications Oxidation (M) sites: Oxidized methionine residues.

• Supplementary file 3. Comparison of SUMO sites identified in this study with sites predicted by SUMOsp.

• Supplementary file 4. Annotated list of SUMO targets.

• Supplementary file 5. Protein diagrams mapping SUMO sites and SIMs on protein secondary structure. Upper Track: Red line – SUMOylated lysine, Black line – lysine, Predicted SUMO Interaction Motifs (gray boxes): Dark gray – high threshold, Medium gray – medium threshold, Light gray – low threshold; Middle Track: Blue box - PFAM domains; Lower Track - Protein Secondary Structure: Purple – Globular Domains, Orange-red – Disorder score, Red – coiled-coil, Black – Transmembrane; Horizontal Axis: amino acid residue number

• Transparent reporting form

## Data availability

Proteomics data have been deposited in the PRIDE archive under the accession code PXD012418.

The following dataset was generated:

| Author(s) | Year | Dataset title | Dataset URL | Database and Identifier |
|---|---|---|---|---|
| Bhagwat NR, Hunter N | 2019 | Meiotic yeast SUMO | https://www.ebi.ac.uk/pride/archive/projects/PXD012418 | PRIDE, PXD012418 |

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
