## [Decision Letter]

**Acceptance summary:**

Post-translational modification of proteins by SUMOylation is critical for meiosis. In this paper, the authors define the stage-specific SUMO proteome in synchronised budding yeast cells undergoing meiotic prophase. Their findings reveal dynamic SUMOylation profiles on a broad range of proteins, particularly highlighting a central role for SUMOylation in chromosomal dynamics in meiosis. This comprehensive analysis of protein SUMOylation provides a rich resource that will inspire future work on meiosis and SUMOylation.

**Decision letter after peer review:**

Thank you for submitting your article "Delineation of the SUMO-modified proteome reveals regulatory functions throughout meiosis" for consideration by *eLife*. Your article has been reviewed by five peer reviewers, including Adèle L Marston as the Reviewing Editor and Reviewer #1, and the evaluation has been overseen by Philip Cole as the Senior Editor. The following individuals involved in review of your submission have agreed to reveal their identity: Federico Pelisch (Reviewer #2); Joao Matos (Reviewer #3); Donald S Kirkpatrick (Reviewer #4).

The reviewers have discussed the reviews with one another and the Reviewing Editor has drafted this decision to help you prepare a revised submission.

Summary

The current work provides a time-resolved global analysis of SUMOylation in yeast meiosis. SUMO is known to be important for several different steps of meiosis in different model systems, but the evidence for substrate-specific (or substrate group-specific) modification is very limited. The authors use a two-step isolation strategy – first, on the protein level purifying His6-tagged SUMO modified substrates on a Ni-NTA resin, which after LysC digestion is followed by anti-diGly antibody-based enrichment of SUMOylated peptides – the authors were able to identify a total of 2747 distinct sites corresponding to 775 target proteins. This dataset stems from six different time points covering all the key events prior to meiosis I: G0, S-phase, DSBs, SI (early repair intermediates), dHJ, and crossover formation. Combined with label-free quantitation, this approach revealed dynamic changes of the SUMO landscape. The authors speculate on how the identified SUMOylation events may regulate different aspects of chromosome organization (such as homolog synapsis and segregation) and all the steps of homologous recombination. In the final figure, the authors test the effect of acutely degrading Aos1 E1 enzyme at particular time points in meiosis, providing evidence for the importance of SUMOylation in the early stages of meiosis.

The study is methodologically and scientifically rigorous and identification of context dependent SUMOylation sites represents an important resource. At the same time, novel insights regarding the mechanistic role of sumoylation are limited. Therefore, although this dataset will be of interest to researchers interested in meiosis and/or SUMOylation, the manuscript would benefit from some follow up biology to validate and/or characterize the specific role of one or more novel SUMOylation events in meiosis.

Essential revisions

1) Validation of hits. The authors did not formally validate their mass spectrometry data for any of the identified proteins. This is a shortcoming that should be addressed. In addition to validating temporal changes in conjugation to specific substrates, it would also be helpful to have a general sense of the relative fraction of proteins that are modified at different stages. Presumably this is high for some proteins (Ubc9), but low for others, but the range is not clear from the data presented. At a minimum, the authors should examine SUMOylation on a handful individual proteins by western blotting (total and Ni-NTA purified) at the same time points as the proteomics analysis. This would allow the reader to see the degree of modification and whether any changes are due to changes in SUMO modification or in protein levels (or both). Looking at the graphs, some of these examples could be Ulp1, Ubc9, Zip3, Tof2, Sir3.

2) Functional analysis. While the authors make many very exciting suppositions based on the identified sites and their prevalence, none of these were tested. It would be ideal if the function of at least one candidate modification could be tested/validated. The authors raise many hypotheses to choose from. For instance, the authors make the very interesting observation that from a total of 17 acceptor sites on Zip3 a single site (K441) makes up for 90% of the measured intensity (Figure 3I). They also provide a compelling hypothesis on what this site-specific modification could do, yet abstain to experimentally test it. Having at least one example to showcase how the dataset can be used to learn about the regulation of a specific protein would provide a great added value to the paper.

3) The Aos1-aid experiments are interesting and suggest that de novo SUMOylation is important at different stages in meiosis. However, the authors did not quantify how much SUMOylation is affected in these experiments. This would seem to be an essential point of reference. E1 subunit depletion is only shown for one condition, pre S-phase. It could be that depletion kinetics are not the same for the other conditions. Authors should show the overall (conjugated and free) Smt3 levels for the experiments, which is the important readout to correlate with the effects on meiosis. While the authors rightly point out that they are seeing the results of inhibiting the novo conjugation, clear evidence of the timing of modification or stability of the conjugates is not shown. In essence, if most of the substrates with meiotic roles are actually long-lived, then acutely depleting the E1 will not reveal their role.

4) The auxin-induced degradation of Aos1-AID and Uba2-AID experiments are lacking a control for effects of auxin in control cells (cells where Aos1 or Uba2 are not degraded i.e. Aos1-AID and Uba2-AID in the absence of auxin). This is necessary to show that the depletion is acute and that protein function is not altered in meiosis by the AID tag.

5) What is the evidence that 6His-Smc3-I96K reports on endogenous SUMOylation? Could the addition of an additional lysine into the SUMO protein alter SUMOylation homeostasis globally? The authors should show a functional test, for example spore viability with Smc3-I96K as the only expressed copy. This is particularly relevant in light of the interesting finding by the authors indicating that most SUMO-modified sites do not conform to the typical Ubc9 binding motif. Furthermore, previous work has already suggested that the Smt3p-I96K mutant is not conjugated to substrates in vivo with the same efficiency as wild type protein (Wholschlegel et al., 2006) so this is a real concern.

6) The strategy of adapting the diGly-IP (initially used with trypsin for Ub and NEDD8) to SUMO, by using a Lys and the consequent use of LysC or LysC+GluC instead of Trypsin has been used in the past (Tammsalu 2014 and papers after this one). The authors should make clear throughout the manuscript that the novelty here is applying a previously developed strategy to meiosis, rather than the strategy itself being novel.

7) The manuscript would easier to digest if it were shortened to focus on a small number of key concepts, rather than an exhaustive discussion/speculation of a large number of targets identified. The authors should also avoid over-speculation where their proposals are not yet backed up by experimental evidence.

8) Since the authors did not measure protein abundance, it is entirely possible that the changes in the pattern of SUMOylation of a given protein are simply determined by the expression level, rather than regulated modification. This is probably the case for many of the proteins detected. It would be worth briefly discussing in the manuscript how specific patterns of SUMO modification might be achieved. Perhaps some of the strongest SUMO "hits" are detected simply because they are the most abundant proteins in the cell, while in reality only a relatively tiny fraction of the total protein is SUMOylated?

9) The authors refer to the dataset as the “SUMO-modified meiotic proteome" (for example in the running title). This can create the wrong impression that the whole of meiosis is analysed – this is not the case as many of the key events in meiosis are not covered (meiosis I, meiosis II, sporulation, etc). The authors should be more precise whenever referring to the dataset as being the “SUMO-modified meiotic proteome”.

---

## [Author Response]

Essential revisions1) Validation of hits. The authors did not formally validate their mass spectrometry data for any of the identified proteins. This is a shortcoming that should be addressed. In addition to validating temporal changes in conjugation to specific substrates, it would also be helpful to have a general sense of the relative fraction of proteins that are modified at different stages. Presumably this is high for some proteins (Ubc9), but low for others, but the range is not clear from the data presented. At a minimum, the authors should examine SUMOylation on a handful individual proteins by western blotting (total and Ni-NTA purified) at the same time points as the proteomics analysis. This would allow the reader to see the degree of modification and whether any changes are due to changes in SUMO modification or in protein levels (or both). Looking at the graphs, some of these examples could be Ulp1, Ubc9, Zip3, Tof2, Sir3.

As requested, in Figure 2—figure supplement 3, we show Western analysis of PCNA, Rad52, Smt3 and Red1. This is accompanied by the following text: “SUMOylation profiles are a compound readout of changes in SUMO modifications and protein levels. The SUMOylation profiles of selected conjugates were validated by Western blot (Figure 2—figure supplement 3); in general, the results matched the proteomics profiles implying that the inferred dynamics tend to reflect changes in target SUMOylation and not simply protein levels (Figure 2—figure supplement 3, A-C). Red1 was a notable exception for which relative SUMOylation levels appeared more or less constant, with changes primarily reflecting the total protein level, which increased throughout prophase I (Figure 2—figure supplement 3D).”

In general, detection of SUMOylated forms requires an IP-Western to be probed with anti-SUMO antibodies, i.e. SUMOylated forms are not readily detected by straight Western analysis. A such, the degree of modification for any given target is very difficult to assess.

2) Functional analysis. While the authors make many very exciting suppositions based on the identified sites and their prevalence, none of these were tested. It would be ideal if the function of at least one candidate modification could be tested/validated. The authors raise many hypotheses to choose from. For instance, the authors make the very interesting observation that from a total of 17 acceptor sites on Zip3 a single site (K441) makes up for 90% of the measured intensity (Figure 3I). They also provide a compelling hypothesis on what this site-specific modification could do, yet abstain to experimentally test it. Having at least one example to showcase how the dataset can be used to learn about the regulation of a specific protein would provide a great added value to the paper.

The following considerations will make it clear why this is not reasonable or necessary:

i) The extensive functional analysis already included in the paper and now presented earlier, in Figure 3, perfectly complements our global SUMO proteomics, confirming that SUMO functions throughout meiotic prophase-I. Moreover, this unique execution-point analysis reveals novel meiotic roles for SUMO, such as DSB formation and the formation and resolution of joint molecules.

ii) Several targets identified by our proteomic analysis have already been studied in some detail in the literature, validating both target identity and the importance of specific sites that we found to be the most abundantly modified. Notably: (a) Ecm11, "20 of the 21 conjugation sites in Ecm11 mapped throughout the predicted unstructured N-terminal region, but SUMOylation at K5 was over four orders of magnitude more intense than at any other site (Figure 5M and Supplementary file 3). K5 was previously shown to be responsible for most Ecm11 SUMOylation and is essential for its function, underscoring the value of our LFQ analysis for identifying functionally relevant conjugation sites (Humphryes et al., 2013; Zavec et al., 2008)"; (b) Ubc9, "15/17 lysines in Ubc9 were identified as SUMO acceptor sites (Figure 3C). K153 accounted for the vast majority of the signal with a 365-fold higher intensity than the next most abundant site. Klug et al. showed that K153-SUMO inhibits conjugase activity and converts Ubc9 into a cofactor that enhances chain formation by unmodified Ubc9 (Klug et al., 2013). This activity is particularly important during meiosis where it facilitates SC formation.”; (c) Red1, "In Red1, a remarkable 32 conjugated sites were identified, 20 of which clustered into a broad domain (492-702) surrounding a lysine-rich patch previously shown to be important for Red1 SUMOylation, SC assembly and crossover interference (Figure 5J and Supplementary file 3)(Eichinger and Jentsch, 2010; Zhang et al., 2014).These include sites that overlap regions shown to interact with Mec3 and Ddc1, components of the 9-1-1 checkpoint-sensor clamp (Eichinger and Jentsch, 2010; Zhang et al., 2014), as well as two SUMO-interaction motifs inferred to mediate interaction with the SC transverse-filament protein, Zip1 (Lin et al., 2010). Additional sites were located at the Red1 closure motif, which is bound by Hop1; and in the C-terminal coiled-coil motif that mediates Red1 oligomerization (West et al., 2019; West et al., 2018). Thus, SUMO may influence all aspects of Red1 function.”

iii) Rigorous, unambiguous functional analysis of any single target represents years of work and a paper’s worth of data. To illustrate this point, we have attached a recently completed manuscript describing the role of Msh4 SUMOylation. Similar studies of Mer2 and Hop1 SUMOylation are ongoing, revealing essential roles in DSB formation. It’s really too much to request that such a study be added to an already large data-rich paper, especially in light of points (i) and (ii) that provide ample functional analysis.

3) The Aos1-aid experiments are interesting and suggest that de novo SUMOylation is important at different stages in meiosis. However, the authors did not quantify how much SUMOylation is affected in these experiments. This would seem to be an essential point of reference. E1 subunit depletion is only shown for one condition, pre S-phase. It could be that depletion kinetics are not the same for the other conditions.

We analyzed E1 depletion by Westerns analysis for all experiments. These are shown in Figure 4—figure supplement 1.

Authors should show the overall (conjugated and free) Smt3 levels for the experiments, which is the important readout to correlate with the effects on meiosis. While the authors rightly point out that they are seeing the results of inhibiting the novo conjugation, clear evidence of the timing of modification or stability of the conjugates is not shown. In essence, if most of the substrates with meiotic roles are actually long-lived, then acutely depleting the E1 will not reveal their role.

We don’t think we’d learn anything from such analysis and we’re unsure why this was deemed an essential revision. Let’s say, for a given condition, we estimate a 60% reduction in SUMO conjugates following Aos1/Uba2 degradation, what does this tell us? What if it’s only a 40% reduction – is the experiment invalid? Given that the E1 is efficiently degraded in each case, any such variation is beyond our control. The point is that we detect strong and distinct phenotypes following E1 depletion at each of the transitions analyzed. Obviously, the types and severity of the phenotypes observed will be a function of extant SUMO conjugates and the lifespans of their modification, but also of the importance of SUMOylation for the function of any given target. Thus, knowing global SUMO levels these experiments will not be informative.

4) The auxin-induced degradation of Aos1-AID and Uba2-AID experiments are lacking a control for effects of auxin in control cells (cells where Aos1 or Uba2 are not degraded i.e. Aos1-AID and Uba2-AID in the absence of auxin). This is necessary to show that the depletion is acute and that protein function is not altered in meiosis by the AID tag.

All experiments in Figure 3 and Figure 3—figure supplement 2 include Aos1-AID / Uba2-AID “no auxin” controls; in fact, in each case, a single culture is split with auxin added to one subculture, as described in the text, “In each case, meiotic cultures were split at the appropriate time point, and auxin was added to one sub-culture to induce degradation of Aos1-AID with the other sub-culture serving as a positive control.”

5) What is the evidence that 6His-Smc3-I96K reports on endogenous SUMOylation? Could the addition of an additional lysine into the SUMO protein alter SUMOylation homeostasis globally? The authors should show a functional test, for example spore viability with Smc3-I96K as the only expressed copy. This is particularly relevant in light of the interesting finding by the authors indicating that most SUMO-modified sites do not conform to the typical Ubc9 binding motif. Furthermore, previous work has already suggested that the Smt3p-I96K mutant is not conjugated to substrates in vivo with the same efficiency as wild type protein (Wholschlegel et al., 2006) so this is a real concern.

*6His-SMT3-I96K/6His-SMT3-I96K* homozygous strains have 96% spore viability. Also, data in Figure 1 and Figure 1—figure supplement 1 show that major events of meiosis remain timely and efficient in this strain. We have clarified this in the text, “functionality of this construct is reflected in the 96% spore viability of *6His-Smt3-I96K/6His-Smt3-I96K* homozygotes and the timing and efficiency of meiosis in these strains, Figure 1B–D and Figure 1—figure supplement 1”

6) The strategy of adapting the diGly-IP (initially used with trypsin for Ub and NEDD8) to SUMO, by using a Lys and the consequent use of LysC or LysC+GluC instead of Trypsin has been used in the past (Tammsalu 2014 and papers after this one). The authors should make clear throughout the manuscript that the novelty here is applying a previously developed strategy to meiosis, rather than the strategy itself being novel.

We made no claim of novelty for this method and cited the original methods papers from the Yates (Wohlschlegel et al., 2006) and Jaffrey (Xu et al., 2010) labs. We have also added a citation for Tammsalu et al., 2015.

7) The manuscript would easier to digest if it were shortened to focus on a small number of key concepts, rather than an exhaustive discussion/speculation of a large number of targets identified. The authors should also avoid over-speculation where their proposals are not yet backed up by experimental evidence.

Our overarching goal for this project was to avoid publishing yet another dry, under analyzed and hard to access proteomics dataset. To this end, we complemented our global proteomics with the long-wave functional analysis of SUMO function, presented in Figure 3, and the comprehensive literature analysis and discussion of identified targets that are most pertinent for meiotic prophase. These efforts bring the dataset to life and provides a rich resource that dramatically lowers the bar for our colleagues to analyze the roles of SUMO for their favorite proteins, providing them with a synthesis of pertinent literature and testable hypotheses.

8) Since the authors did not measure protein abundance, it is entirely possible that the changes in the pattern of SUMOylation of a given protein are simply determined by the expression level, rather than regulated modification. This is probably the case for many of the proteins detected. It would be worth briefly discussing in the manuscript how specific patterns of SUMO modification might be achieved. Perhaps some of the strongest SUMO "hits" are detected simply because they are the most abundant proteins in the cell, while in reality only a relatively tiny fraction of the total protein is SUMOylated?

Please see our response to point 1. This is more or less the same issue.

9) The authors refer to the dataset as the “SUMO-modified meiotic proteome" (for example in the running title). This can create the wrong impression that the whole of meiosis is analysed – this is not the case as many of the key events in meiosis are not covered (meiosis I, meiosis II, sporulation, etc). The authors should be more precise whenever referring to the dataset as being the “SUMO-modified meiotic proteome”.

We made the title, running head and subheading less specific.